

**Ice core records of levoglucosan, dehydroabietic and vanillic acids from Aurora**

**Peak in Alaska since the 1660s: A proxy signal of biomass burning activities in**

**the North Pacific Rim**

Ambarish Pokhrel[1,2,3], Kimitaka Kawamura[1,2,*], Kaori Ono[1], Akane Tsushima[1],

Osamu Seki[1], Sumio Matoba[1], Takayuki Shiraiwa[1,] and Bhagawati Kunwar[2]

[1]Institute of Low Temperature Science, Hokkaido University, Sapporo, Japan

[2] Chubu Institute for Advanced Studies, Chubu University, Kasugai, Japan

[3]Asian Research Center, Kathmandu, Nepal

[*]Corresponding author

Email address: kkawamura@isc.chubu.ac.jp (K. Kawamura)

Revision to Atmospheric Chemistry and Physics (ACP 2019-139)

## Abstract

A 180 m-long (343 years) ice core was drilled in the saddle of Aurora Peak in Alaska (63.52°N; 146.54°W, elevation: 2,825 m) and studied for biomass burning tracers. Concentrations of levoglucosan and dehydroabietic and vanillic acids exhibit multi-decadal variability with higher spikes in the 1678, 1692, 1695, 1716, 1750, 1764, 1756, 1834, 1898, 1913, 1966 and 2005 A.D. Historical trends of these compounds showed enhanced biomass burning activities in the deciduous broad leave forests, boreal conifer forests and/or tundra woodland and mountain ecosystems before the 1830s and after the Great Pacific Climate Shift (GPCS). The gradually elevated level of dehydroabietic acid after the GPCS is similar to p-hydroxybenzoic acid (p-HBA) from Svalbard ice core, suggesting common climate variability in the Northern Hemisphere. The periodic cycle of levoglucosan, which seemed to be associated with the Pacific Decadal Oscillation (PDO), may be more involved with the long-range atmospheric transport than other species. These compounds showed significant correlations with global lower tropospheric temperature anomalies (GLTTA). The relations of the biomass burning tracers with PDO and GLTTA in this study suggest that their emission, frequency, and deposition are controlled by the climate driven forces. In addition, historical trends of dehydroabietic and vanillic acids (burning products of resin and lignin, respectively) from our ice core demonstrate the northern hemispheric connections to the common source regions as suggested from other ice core studies from Svalbard, Akademii Nauk and Tunu Greenland in the Northern Hemisphere.

(Words: 240)

## 1. Introduction

Biomass burning tracers (e.g., levoglucosan, dehydroabietic, vanillic, p-droxybenzoic, and syringic acids) are ubiquitous in the atmosphere and well deposited on ice sheets as snow particles (i.e., precipitation) (Muller-Tautges et al., 2016; Grieman et al., 2018a,b; Shi et al., 2019). Previously, ammonium ($NH_4^+$), nitrite ($NO_2^-$), nitrate ($NO_3^-$) and sulfate ($SO_4^{2-}$) were used to understand the atmospheric signals of biomass burning and/or the Pioneer Agriculture Revolution (PIA-GREV) in the Northern Hemisphere (Holdsworth et al., 1996; Legrand and Mayewski, 1997; Legrand et al., 2016). For instance, a signal of biomass burning is ammonium (e.g., $[NH_4]_2SO_4$) in snow particles, which is a constituent of forest fire smoke (Holdsworth et al., 1996; Tsai et al., 2013). Biomass burning activities such as forest fires and residential heating may affect climate change (Legrand and De Angelis, 1996; Savarino and Legrand 1998; Gambaro et al., 2008; Keywood et al., 2011).

Ice core records archive the long-term changes in deposition and concentration of organic (e.g., biomass burning tracers, ethane, formate, acetate, dicarboxylic acids, pyruvic acid and $\alpha$-dicarbonyls) and inorganic species (e.g., $NH_3^+$, $SO_4^{2-}$, $NO_3^-$, $K^+$ and $NO_2^-$) (Yang et al., 1995; Legrand and Mayewski, 1997; Andreae and Merlet, 2001; Kaufmann et al., 2010; Lamarque et al., 2010; Wolff et al., 2012; Kawamura et al., 2012; Kehrwald et al., 2012; Legrand et al., 2016; Shi et al., 2019). Many studies have shown that there are some discrepancies of temporal and spatial biomass burning activities in ice core proximity records (Legrand et al., 1992,1996; Kaplan et al., 2010; Kawamura et al., 2012; Grieman et al., 2015; Rubino et al., 2016; Legrand et al., 2016; Grieman et al., 2017; Zennaro et al., 2018; Li et al., 2018; Grieman et al., 2018a,b; You et al., 2019) in both Northern and Southern Hemisphere (NH/SH).

Previous proxy records of biomass burning activities from Lomonosovfona, Svalbard (Grieman et al., 2018a) showed different trend between vanillic acid and p-hydroxybenzoic acid (p-HBA) within the same ice core sample. Interestingly, ice core records of NEEM (Zennarao et al., 2014; 2018) demonstrated a human impact on the climate system since four thousand years ago. A different circumpolar region in the NH has a different atmospheric airmass circulation with different results of biomass burning tracers such as levoglucosan, vanillic, dehydroabietic and syringic acids, ethane, ammonium and other carboxylic acids, suggesting potential discrepancies of origin, transport, and deposition of these compounds on the ice crystals.

These discrepancies of biomass burning tracers in different ice core records may suggest the different glacio-chemical cycles in the NH and SH throughout decadal to centennial and even millennia. For example, centennial and/or shorter time scale of trends exhibited different elevated/suppressed concentration trends of p-HBA/vanillic acid during 1600 A.D. and vanillic/p-HBA during 2000-2008 A.D. (Grieman et al., 2018a). Similarly, Svalbard ice core record (Grieman et al., 2018a) showed different elevated/suppressed historical trends/peaks from NEEM- ice core of Greenland (Zennaro et al., 2018). These results most likely suggest the occurrence of changing/shifting contributions of source regions with the different ecosystem of trees, shrubs, and grasses to the sampling sites.

There are a few ice core studies of biomass burning-derived specific organic tracers, including levoglucosan that is a pyrolysis product of cellulose and hemicellulose and other sugar compounds such as mannosan and galactosan, as well as dehydroabietic and vanillic acids which are biomass burning products of resin and lignin, respectively (Kawamura et al., 2012; Legrand et al., 2016; Grieman et al., 2017; Zennaro et al., 2018; Li et al., 2018; Grieman et al., 2018a,b; You et al., 2019).

Kawamura et al. (2012) reported specific biomass burning tracers (levoglucosan,
dehydroabietic and vanillic acid) for an ice core (1693-1997 A.D.), collected from the
Kamchatka Peninsula (56º04'N, 160º28'E, Elevation: 3,903 m) in the western North
Pacific Rim.

In this paper, we report levoglucosan, dehydroabietic acid and vanillic acid in

an ice core collected from Aurora Peak of southern Alaska, an inland site facing to the
northeast of Pacific Ocean. This ice core covers 1665-2008 A.D., which can help to
better understand the historical variability in the atmospheric transport of biomass
burning tracers between the western North Pacific (Kawamura et al., 2012) and
eastern North Pacific (this study). We also compare the present results with other ice
core studies from Greenland, Svalbard and Akademii Nauk in the NH. The results of
this study can further disclose the database of levoglucosan, dehydroabietic and
vanillic acids from the alpine glacier in the North Pacific Rim to explore their possible
sources, origin, long- and short-range atmospheric transport, ecological changes and
climate variability in the NH.
**2. Materials and Methods**

An ice core (180.17 m deep, 343 years old) was drilled in the saddle of the

Aurora Peak of southern Alaska (location: 63.52°N, 146.54°W, elevation: 2,825 m,
see Fig. 1 for sampling site). The annual mean temperature at the site was minus
2.2°C, which matched to the temperature of 10 m depth in the borehole-ice. The
annual accumulation rate of snow is 8 mm $yr^{-1}$ since 19 century and 23 mm $yr^{-1}$ after
the Great Pacific Climate Shift (GPCS, cold water masses were replaced by warm
water since 1977, e.g., Meehl et al, 2009). The 180 m long core was divided into ~50
cm pieces and directly transported to the laboratory of the Institute of Low
Temperature Science, Hokkaido University, Japan and stored in a dark, cold room at -
20ºC until analysis.

The ice core ages were determined by using annual counting of hydrogen

isotopes (δD) and Na$^+$ seasonal cycles (Tshushima, 2015; Tsushima et al., 2015) with
tritium-peak reference horizons of 1963 and 1964 and volcanic eruptions of Mt. Spurr
and Mt. Katmai in 1992 and 1912 with dating error was ±3 years of 0.02 m resolution.
These ice core samples (50 cm long, one-quarter cut by circumference) were
mechanically shaved off (~5 – 10 mm thickness of the out core surface) on a clean
bench at -15°C in a cold room. A ceramic knife was used to avoid a possible
contamination during sample collection. We cleaned ceramic knife (total 12 times)
three times by using organic free pure water (MiliQ water), methanol (MeOH),
dichloromethane (DCM) and a mixture of 2:1 of DCM and MeOH. These scraped ice
samples were placed in a clean glass jar (Iwaki Glass, 1000 mL) for 24 hours with
aluminum foil as a cap cover in a level-2 clean room, After 24 hours, these shaving
ice core samples were kept at room temperature (ca. 25°C) to which small amount (ca.
10 mg) of $HgCl_2$ was added (Pokhrel, 2015). Finally, the thawed sample was
transferred into a 800 ml pre-cleaned brown glass bottle and stored at 4°C. The clean
glass jars and bottles were pre-heated at 450°C for 12 hours. The total number of ice
core sections was 147 with sampling frequency of ~40% of ice core.

These melted ice core samples (150 mL) were concentrated to almost dryness

using a rotary evaporator under a vacuum in a pear shaped flask (300 ml) and
extracted by a mixture of DCM/MeOH (2:1) using an ultrasonic bath. The extracts
were transferred to 1.5 mL glass vial and dried under a nitrogen stream. Extracts were
derivatized with 99% N, O-bis-(trimethylsilyl)trifluoroacetamide (BSTFA) + 1%
trimethylchlorosilane (TMCS) and 10 μl of pyridine at 70°C for three hours (Fu et al.,
2011; Kawamura et al., 2012). Before injection to gas chromatography (GC)/mass
spectrometry (MS), known volume of internal standard (n-$C_{13}$ alkane) was added. GC
peaks were analyzed by GC/MS: a Hewlett–Packard Model 5973 MSD coupled to a
HP 6890 GC using a capillary column (HP-5MS, 30 m×0.32 mm I.D., 0.25 μm film
thickness) installed with a split/splitless injector. The GC oven temperature was
programmed from 50°C (2 min) to 120°C at 30°C/min, and then to 300°C at 6°C/min
and maintained at 300°C for 20 min. Helium was used as a carrier gas. The GC/MS
was operated on a scan mode (m/z=50-650) with an electron impact mode at 70 eV
(Pokhrel et al., 2016).
Fragment ions at m/z = 217, 204 and 333 for levoglucosan, m/z = 239 for
dehydroabietic acid and m/z = 297/312/267 for vanillic acid were processed on the
Chemistation software and used for quantification. Peaks were further confirmed by
comparing the mass spectra with those of authentic standards and the mass spectral
data in the NIST/Willey library. An aliquot of authentic standard solution (10 μL)
containing levoglucosan, dehydroabietic acid, vanillic acid and syringic acid (5.5, 4.7
and 4.2 ng/μL, respectively) was spiked to organic free Milli-Q water (200 ml) placed
in the pear-shaped flask. The water sample was concentrated and dried by the
procedure described above. The concentrates were derivatized with BSTFA and peaks
were analyzed by GC/MS. The recoveries of the spiked samples of levoglucosan,
dehydroabietic acid, and vanillic acid were more than 83%. Duplicate analyses were
conducted to check analytical error of target compounds, which were less than 9%.
Laboratory blanks was measured using Milli-Q water (200 ml). The procedural blanks
showed no detectable peaks of these compounds. Detection limits of these species
were 0.002-0.005 ng/kg-ice.
We performed cluster analysis for 10 days backward trajectories at 500 hPa
for 2002 to 2007 (Fig. 7a-f) computed for every 6 hours, which disclose that long-
range atmospheric circulation was significant in the study site of Aurora Peak. To
identify the possible source regions of biomass burning events, we prepared annual
composite maps (2002 to 2008 AD) of the Moderate Resolution Imaging
Spectroradiometer (i.e. MODIS) effective hot spot (Fig. 7a-f) from the Earth
Observing System Data and Information System (EOSDIS) using the Terra and Aqua
satellites of NASA (https://earthdata.nasa.gov/data/near-real-time-data/firms/active-
fire-data). The 10 days backward trajectory analysis from 2002 to 2007 showed that
Aurora Peak received air masses from the North Pacific Ocean, East Asia, Siberia,
Europe, Canada, and higher latitude of Alaska (Fig. 7a-f). Similar sources were
reported using 10-days backward trajectory from 1992-2002 (>300 hPa) (Yasunari
and Yamazaki, 2009). The Kamchatka Peninsula also receives air masses from China,
Mongolia, Siberia, Eastern Russia, and Europe (Kawamura et al., 2012).
**3.  Results and Discussion**
Anhydrosugars such as levoglucosan are ubiquitous in the atmosphere, which
are emitted significantly from biomass burning activities and deposited on the ice
crystals, and contribute to water-soluble organic carbon (WSOC) (Gennaro et al.,
2015; Verma et al., 2015; Gao et al., 2015; Legrand et al., 2016; Grieman et al., 2017;
Li et al., 2018; Grieman et al., 2018a,b; You et al., 2019). These are produced from
the pyrolysis and combustion of cellulose and/or hemicellulose from wildfires and
domestic wood fires at temperatures above 300 °C (Shafizadeh, 1984; Fraser and
Lakshmanan, 2000; Simoneit et al., 2002) during the smoldering stage of a fire.
Recently, Kuo et al. (2011) reported that levoglucosan and its isomers are produced at
temperature up to 350°C. Many studies have shown that levoglucosan is the most
abundant anhydrous monosaccharide (Engling et al., 2006; Hoffmann et al., 2010;
Kuo et al., 2011), which is tracked by other species of anhydromonosaccharides; e.g.
mannosan, galactosan and/or dehydroabietic acid. Such specific characters and the
sources can make levoglucosan a unique tracer (Simoneit et al., 1999; Jordan et al.,
2006) in the southern Alaska as shown in Fig. 1.
In addition, historical trends of biomass burning tracers may represent the bulk
effects of emissions, transport, transformations, and depositional and post-
depositional process on the ice crystals (Grieman et al., 2017). Comparison of this
study (e.g., concentration trends) with other ice core studies suggested that these
compounds are well captured in the atmosphere and deposited to the ice sheets.
Backward trajectories of this study and other ice core studies suggested common
source regions (e.g., Russia, Siberia, and East Asia), from which it takes several days
to reach the sampling sites (e.g., Greenland-Tunu, Svalbard, Akademii Nauk, and
Aurora Peak of Alaska).
**3.1 Levoglucosan**
This study showed that average concentration of levoglucosan (range: BDL-
20800, average: 543±2340 ng/kg-ice) is 8.6 times higher than that of dehydroabietic
acid (range: BDL-556, ave. 62±97 ng/kg-ice) and 400 times higher than that of
vanillic acid (range: BDL-18.6, ave. 1.5±2.9 ng/kg-ice) for 1665-2008 A.D. It should
be noted that combustion of lignite (lignite includes fossilized cellulose) or
hemicellulose emits levoglucosan and its isomers; e.g., mannosan and galactosan
(Hoffmann et al., 2010; Kuo et al., 2011). However, we did not detect these isomer
compounds (less than DL). In contrast, higher concentrations of these isomers and
levoglucosan were reported in aerosol samples collected from the oceans via "round-
the-world cruise" (Fu et al., 2011), Mt. Tai in the North China Plain (Fu et al., 2008),
and urban tropical India (Fu et al., 2010) using the same method.
Levoglucosan may not be as stable as previously thought in the atmosphere
(Fraser and Lakshmanan, 2000; Hoffmann et al., 2010), however, its concentrations
are not seriously influenced during transport for several days (Fraser and Lakshmanan,
2000; Lai et al., 2014). Hence, we may speculate that levoglucosan could be stable
enough in the ice core studies. However, degradation of levoglucosan depends upon
the OH radical (Hennigan et al., 2010), which are automatically affected by relative
humidity of the atmosphere and air mass aging during long range atmospheric
transport from Japan, China, Mongolia, Siberia, and Russia to Aurora Peak.
Levoglucosan showed higher concentrations in around 1660s-1830s (Figure
2a) with sporadic peaks in 1678 (ice core depth: 174.6 m; concentration: 593 ng/kg-
ice), 1692 (172.2 m; 704), 1695 (170.3 m; 1250), 1716 (165.6 m; 990), 1750 (156.7
m; 913), 1764 (151.5 m; 1433), 1786 (147.3 m; 7057), 1794 (146.1 m; 3302) and
1834 (138.4m; 944) above its average concentration (542 ng/kg-ice). Source regions
of these higher spikes could be East Asia, Eastern Russia, Siberia, higher latitudes of
Alaskan regions, and Canadian regions. For instance, Ivanova et al. (2010) reported
the frequently occurred heavy forest fires (e.g., boreal forest) in spring, summer and
autumn in eastern Siberia in the past, which is a potential source region to Alaska.
This study showed higher concentrations of levoglucosan before the 1840s (Figure
2a). Marlon et al. (2008) further confirmed that there was intensive biomass burning
between the 1750s -1840s on a global scale, which is linked to increasing
anthropogenic activities (e.g., population growth and land-use change).

Similarly, we detected higher spikes of levoglucosan in 1898 (120.7 m; 577

233        ng/kg-ice), 1913 (114.8 m; 20800), 1966 (77.7 m; 692) and 2005 (13.7 m; 598) above

the average concentration (542). Figure 2a clearly shows its lower levels than the

average after the 1830s (except for 1898, 1913, 1966 and 2005 A.D.) compared to

before 1830s. This decline could be attributed to less forest fire activity due to

intensive grazing, agriculture, and forest fire management system (Marlon et al., 2008;

Eichler et al., 2011). It should be noted that charcoal signals are scarce for Siberian

regions compared to North American and European ice core records (Eichler et al.,

2011). Moreover, two-third of Earth's boreal forest (17 million $km^2$) lies in Russia,

which is a potential source of forest fires with a significant effect on a global air

quality (Isaev et al., 2002; Eichler et al., 2011).

Ice core records of Mt. Logan from Canada, GISP2 and 20D (older than the

1850s) from Greenland are characterized by higher spikes of $NH_4^+$ superimposed with

relatively uniform summertime and wintertime minimum (Whitlow et al., 1994). We

obtained higher spikes of levoglucosan before the 1840s (Fig. 2a), which is consistent

with higher spikes of $NH_4^+$ in 1770-1790 and 1810-1830 in the Mt. Logan data (e.g.,

Whitlow et al., 1994). This comparison suggests similar source regions of $NH_4^+$ for

different sampling sites before the 1830s. In contrast, Mt. Logan data showed higher

spikes of $NH_4^+$ in the intervals of 1850-1870 and 1930-1980, which is dissimilar

(except for two points) to our results from Aurora Peak (Fig. 2a). It should be noted

that Greenland ice core records (GISP2 and 20D) showed lower spikes of $NH_4^+$

compared to Mt. Logan (Whitlow et al., 1994) during these intervals (1850-1870 and

1930-1980). This is consistent with the results of Aurora Peak (except for 1966),

again suggesting similar source regions (Holdsworth et al., 1992; Davidson et al.,

1993; Whitlow et al., 1994). The potential source regions for Greenland ice cores

include northern North America, Europe, and Siberia, which are also likely source
regions for Mt. Logan (Holdsworth et al., 1992; Davidson et al., 1993; Whitlow et al.,
1994; Legrand et al., 2016). These regions may be associated with higher spikes in ice
cores from Mt. Logan, Greenland and Aurora Peak of Alaska.
Except for a few points, e.g., 1999 (436 ng/kg-ice) and 2005 (598),
concentrations of levoglucosan drastically decreased in 1980-2008. This decrease
infers that forest fire activities could be depressed by many factors. For instance,
Central and East Siberian forest fire activities were controlled by strong climate
periodicity, e.g., Arctic Oscillation (AO), El Nino, intensification of the hydrological
cycle in central Asia, and other human activities in the NH (Robock, 1991; Wallenius
et al., 2005; Balzter et al., 2007; Achard et al., 2008; Eichler et al., 2011). Eichler et al.
(2009) further confirmed that from 1816 to 2001 higher amounts of $NH_4^+$ and formate
(HCOO$^-$) were directly emitted from biogenic sources rather than biomass burning
(Olivier et al., 2006) in the Belukha glacier in the Siberian Altai Mountains. Moreover,
lower concentrations of charcoal between 1700 and 2000 in this Altai Mountain
further suggest that forest fire activities were weaker than anthropogenic activities in
the source regions (Eichler et al., 2011).
Similarly, the sparsity of levoglucosan after the 1840s compared to the period
of 1660s to 1840s means low intensity of biomass burning and/or significant
deposition before reaching to the saddle of Aurora Peak, except for 1898, 1913, 1947
and 1966 A.D., which could be due to a point source around Alaskan region for
levoglucosan rather than long-range atmospheric transport. For example, higher
spikes of $NH_4^+$ at Mt. Logan during 1900-1990 A.D. are likely originated from central
and eastern Siberia (Robock, 1991), which is dissimilar to the source regions in this
study. The only exception is 1966 (2000 ng/kg-ice), suggesting that local biomass
burning and/or different source regions could be activated for levoglucosan is
important in southern Alaska during this period. Moreover, vanillic acid (VA) and p-
hydroxybenzoic acid (p-HBA) of Svalbard and Akademii Nauk (Eurasian Arctic) did
not show similar trends (Grieman et al., 2017, 2018a). It further suggests that central
and eastern Siberian regions did not contribute this compound significantly during
this period (1900-1990 A.D.) compared to other ice core studies (e.g., Fig. 6a-e)
and/or atmospheric circulations could be shifted.
The above results suggest the subsequent evidences: (a) heavy biomass
burning could be activated in the source regions, (b) short-range air mass circulation
could quickly reach southern Alaska, causing higher levels of levoglucosan; (c)
dilution and/or scavenging of biomass plume enroute could be maximized after 1830s,
whose mechanisms could be associated with dry and wet deposition, diffusion, and
degradation by hydroxyl radicals in the atmosphere during long range atmospheric
transport, (d) a common NH summertime biomass burning plume could be
significantly deposited during short-range atmospheric circulation on the exposed
surface area of the glaciers. Particulary, Mt. Logan, Svalbard, Tunu of Greenland and
Aurora have common source regions, e.g., Russia and/or Siberian forest as well North
America/Canadian forest (Figure 6a-e). These considerations support that Alaskan
glaciers can preserve most biomass burning events in the circumpolar regions, which
occurred in the source regions of Siberia, East Asia, Canada and Alaska.
Hence, these historical records of levoglucosan before the 1830s suggest that
long-range atmospheric transport was significant rather than short-range transport
from intense and widespread forest fires. For instance, forest fire intensity in 1660s-
1830s A.D. could be induced by lightning during drought seasons in the Siberian
regions as well as extensive burning to clear land for agriculture purposes in the NH
(Whitlow et al., 1994; Legrand et al., 2016; Grieman et al., 2017; 2018a, b).

A declining trend in the concentrations of levoglucosan after the 1830s (except

for few points) showed that sources could be changed significantly and/or forest fire
activities could be suppressed and/or controlled in 1830s-1980s (Whitlow et al., 1994).
It should be noted that 1400 A.D. to the end of the 1700s A.D. is the Little Ice Age
(LIA) and after LIA to late 1800s is considered as the extended Little Ice Age (ELIA)
(Mann et al., 2009; Divine et al., 2011;). This study shows that intense biomass
burning activities (higher spikes) before the 1830s are somewhat similar to historical
records of p-HBA and vanillic acid of Lomonosovfonna (Svalbard) and Akademii-
Nauk ice core in the NH (Grieman et al., 2017, 2018a) except for some points (Fig.
6a,b,d). Hence, recent changes in the concentration trends in the Alaskan ice core are
thought to be climate-driven. These climate-driven effects are further discussed in
later section 3.4.
**3.2 Dehydroabietic acid**

Dehydroabietic acid is produced by pyrolytic dehydration of abietic acid from

conifer resin. In other words, dehydroabietic acid is produced during the burning
process of conifer resins (Simoneit et al., 1993; Kawamura et al., 2012;). It can be
used as a specific biomass-burning tracer for conifer trees and other resin-containing
softwoods in an ice core study. Dehydroabietic acid was detected as the second
dominant species (range: BDL-556, ave. 62.4±97.2 ng/kg-ice), whose concentrations
are 9 times lower than levoglucosan but more than 46 times higher than vanillic acid
(range: BDL-18.6, ave. 1.62±2.96 ng/kg-ice). Dehydroabietic acid showed higher
spikes in 1678 A.D. (ice core depth in meter, 173.9 m; 200 ng/kg-ice), 1716 (165.3 m;

67.5 ng/Kg-ice), 1728 (161.5 m; 139 ng/Kg-ice), 1732 (159.6 m; 233 ng/Kg-ice), 1738 (158.3 m; 113 ng/Kg-ice), 1750 (156.7 m; 66.9 ng/Kg-ice), 1764 (151.5 m; 331 ng/Kg-ice), 1786 (147.3 m; 386 ng/Kg-ice), 1794 (146.1 m; 78.6 ng/Kg-ice), 1913 (114.8 m; 101 ng/Kg-ice) than its average concentration (62.4 ng/kg-ice), and each consecutive years from 1994 to 2007 A.D. (depth range: 44.8-0.88 m) have concentrations of 92.8, 199, 141, 203, 136, 109, 98.5, 124, 124, 174, 309, 131, 298, and 555 ng/kg-ice. Vanillic acid from Svalbard (Grieman et al., 2018a) showed similar spikes with dehydroabietic acid in this study during the 1660s to 1790s A.D. In addition, Svalbard ice core showed relatively lower spikes from 1800s to 1980s as compared to 1660s-1790s A.D. In contrast, p-HBA in this study did not show a similar trend with Svalbard (Fig. 6a,b).

These periods are consistent with the higher spikes of levoglucosan, except for a few points (e.g., 1734-1738 A.D.) before 1990 A.D. (Fig. 2a, b). The historical trend of dehydroabietic acid is also similar to that of levoglucosan before 1980, which is consistent with Kamchatka ice core records (Kawamura et al., 2012). In contrast, Kamchatka ice core showed a gradual increase of dehydroabietic acid after the 1950s. However, we found an abrupt increase for dehydroabietic and vanillic acids in the Alaskan ice core after 1980 A.D. (Fig. 2b,c). These results suggest that biomass burning plumes of pine, larch, spruce and fir trees in Siberian regions have a substantial influence on Kamchatka, southeast Russia (facing to the western North Pacific Rim) than southern Alaska (facing to the eastern North Pacific Rim).

We found that concentrations of dehydroabietic acid in the Alaskan ice core after the 1980s were higher than those of levoglucosan, which is consistent with Kamchatka records (Kawamura et al., 2012). This further suggests that biomass burning plumes from Siberian boreal conifer trees could be transported to the North

Pacific regions including the eastern North Pacific Rim. It also suggests that East
Asian regions (broad-leaf trees are common) could be important for levoglucosan
rather than dehydroabietic acid (boreal forest fires in Siberia, where pine trees are
dominant). For instance, correlation of levoglucosan versus dehydroabietic and
vanillic acid from 1660 to 1840 are weak but significant ($\tau=0.37$ and 0.33, $p<0.05$,
respectively), suggesting the presence of common source region. Correlation of
levoglucosan with dehydroabietic and vanillic acids from 1920 to 1977 are not
significant (0.11 and 0.14, respectively). On the other hand, vanillic vs.
dehydroabietic acid showed significant correlation (0.41, $p<0.01$), suggesting a
different source region for levoglucosan. Backward trajectories analysis (500 hPa) of
air masses (2002-2007 A.D.) together with fire counts also showed that sources
regions also include Mongolia, China and Japan (Fig. 7a-f). Yasunari and Yamazaki
(2009) reported that Alaska can receive air masses from East Asia and Japan in the
troposphere (>300 hPa). The Kamchatka Peninsula also can receive air masses from
these regions (Kawamura et al., 2012).

These results showed some similarity in the records of levoglucosan between

Kamchatka and Alaska ice cores (except for few points) and some discrepancies of
dehydroabietic acid between two sampling sites. Dehydroabietic acid concentrations
gradually increased in the Kamchatka ice core after the 1950s. Alaskan ice core
showed an increase after the 1970s (Fig. 6e), suggesting that conifer-burning plumes
could be transported significantly to Kamchatka as well, but not southern Alaska in
the 1950s-1980s. There is another possibility for this discrepancy between two sites,
i.e., dehydroabietic acid could be decomposed during long-range atmospheric
transport (Simoneit and Elias, 2001) from Siberia to southern Alaska although it could
easily reach to Kamchatka in the western North Pacific Rim. The Kamchatka ice core
also did not show high spikes (except 1970) in the 1950s-1970s. Such types of lower
spikes and/or sporadic peaks of levoglucosan and dehydroabietic acid after the 1910s
(Fig. 2a,b) and the correlations suggest that source regions should be different (e.g.
East Asian broad leaf trees and Siberian boreal forest/pine trees), or regional transport
overwhelms the long range atmospheric transport of dehydroabietic acid rather than
levoglucosan over the saddle of Aurora Peak at least after the 1910s. Interestingly,
dehydroabeitic acid showed an increasing trend from 1980s to onwards with higher
concentrations than levoglucosan, being consistent with Kamchatka ice core
(Kawamura et al., 2012).

Annual composite maps (Fig. 7a-f) of the Moderate Resolution Imaging

Spectroradiometer (MODIS) from 2001 to 2007 show a continental outflow of air
masses from Eurasia to the Aurora site, generally supporting the above results and
implications for the Alaskan ice core. However, we detected higher spikes of
levoglucosan (in 2004, 2005 and 2006 A.D. with 95, 598 and 131 ng/kg-ice,
respectively), dehydroabietic acid (in 2004, 2006 and 2007 A.D. with 309, 298 and
556 ng/kg-ice, respectively) and vanillic acid (in 2005, 2006 and 2007 A.D. with 18.6,
7.30 and 12.7 ng/kg-ice, respectively) within these years, suggesting that they have
different sources. It is well known that 2004 is the year of biomass burning in Alaska.
The concentration of dehydrobaietic acid in 2004 (309 ng/kg-ice) is three times higher
than levoglucosan (95.3 ng/kg, see Fig. 2), suggesting that boreal forest fires
associated with conifer trees followed by short- and long-range atmospheric transport
are more important in recent decades in the Northern Hemisphere.
**3.3 Vanillic acid**

We detected vanillic acid (VA) in the ice core from Aurora Peak (Fig. 2c),

which is a biomass-burning tracer of lignin (Simoneit et al., 1993). Particularly,

vanillic acid can be produced by incomplete combustion of conifer trees, i.e., conifer-

rich boreal forest (Simoneit et al., 1993; Pokhrel, 2015). We found that the levels of

vanillic acid are very low between 1830s and 1960s as shown in Figure 2c. Higher

spikes of a lignin tracer were detected in the following years: 1678 (3.25 ng/kg-ice),

409        1692 (3.23), 1695 (5.56), 1732 (3.98), 1786 (3.60), 1814 (11.0), 1818 (5.50), 1973

410        (5.52), 1989 (3.57), 1993 (2.66), 1996 (4.66), 1997 (3.5), 1999 (3.57), 2001 (3.26),

and 2007 (18.6). We found that the spikes of vanillic acid are not consistent with

those of levoglucosan in the ice core during the periods (Fig. 2). In particular, in more

recent years after 1990, vanillic acid showed a clear abrupt increase in the ice core,

which is consistent with the increase of dehydroabietic acid but different from

levoglucosan (Fig. 2). The abrupt increase of vanillic acid in the Alaskan ice core is

consistent with that of the Kamchatka ice core (Kawamura et al. 2012).

The higher concentrations and similarity of vanillic and dehydroabietic acids

in the Alaskan ice core after the 1990 suggests an enhanced emission of biomass

burning products of conifer trees and lignin in the boreal forests in Alaska, which

could be imprinted in the southern Alaska ice core. Interestingly, we found a

significant correlation (Fig. 3a) between dehydroabietic acid (except for 2005 A.D.)

and vanillic acid ($\tau=0.60$, $p<0.01$) after 1990s, whose period corresponds to the Great

Pacific Climate Shift (GPCS, 1977-2007 A.D.). Being consistent with the warmer sea

surface temperature in the eastern North Pacific Rim during the GPCS periods (Meehl

et al., 2009), southern Alaska is influenced by the warmer temperature and dryness,

which triggered more chance of forest fires in the boreal forests, causing more

emissions of conifer and lignin tracers over the southern Alaskan atmosphere (Figs. 2

and 6). Interestingly, Kamchatka ice core also showed an increased concentration of
these tracers after1970s (Kawamura et al., 2012).

Vanillic acid in the Alaskan ice core showed different trend from Svalbard ice

core (Fig. 6e) after the GPCS (1976-77), suggesting different source regions.
Dehydroabietic acid exhibits similar trend with p-hydroxybenzoic acid (p-HBA) of
Svalbard ice core (Grieman et al., 2018a). p-HBA is produced from tundra grasses
and peat species, suggesting a similar source of North Asia including Siberia. Its ice
core record may be climate-driven in the North Pacific Rim. In contrast, the historical
trend of vanillic acid from the 1770s to 1950s is similar to that (depressed trend) of
Tunu Greenland ice core, except for few years of 1851, 1870, 1880, 1934, and 1946
(Fig. 6c), which infers that long range atmospheric transport from Russia may be a
likely source. These two trends diverge markedly after the 1950s onwards. In addition,
vanillic acid in this study exhibits a similar trend with p-HBA and vanillic acid in the
ice core from Akademii Nauk (Grieman et al., 2017) in 1890s-1980s (Fig. 6d).

These results suggest that Alaskan glacier showed non-stationary multi-

decadal variability of biomass burning tracers from tundra grasses and peat species.
Notably, during the 1660s to 1820s, vanillic acid, dehydroabietic acid, and
levoglucosan have higher spikes (Fig. 6a,b,c) at 4 to 9 points, which are common in
other ice cores (Fig. 6a-d) in the NH. After these higher spikes, global (at least Tunu,
Akademii Nauk and Aurora) depression of vanillic acid and p-HBA (1830s-1950) can
be observed (e.g., Fig. 6a-d) in the NH, suggesting that similarity and variability of
these acids are temporally and spatially heterogeneous in the NH under the climate
driven forces. Historical trends of biomass burning tracers from this and other ice core
studies, together with backward trajectories (Fig. 7a-f), suggest a common potential
source region of North Asia and North America, which are characterized by fire
activities of boreal tundra woodlands, boreal conifer forests and peat.
Dehydroabietic acids and p-HBA may be unstable compared to photo-
degradation of levoglucosan during long-range transport. For instance, a higher
sensitivity of dehydroabietic acid was reported compared to levoglucosan (Simoneit
and Elias, 2001; Simoneit et al., 2002). It should be noted that we did not detect p-
HBA, which can be produced from incomplete combustion of grasses (Simoneit et al.,
2002; Kawamura et al., 2012;) although showed p-HBA was detected in Kamchatka
ice core (Kawamura et al., 2012). In contrast, we detected significant amounts of
dehydroabietic acid from 1665-2007 in this study (Figure 2b). Hence, we may
speculate that p-HBA could be unstable compared to levoglucosan, dehydroabietic
acid and vanillic acid during long-range atmospheric transport.
Moreover, the historical trend of vanillic acid from 1800-2000 in Greenland
ice core (McConnell et al., 2007) is entirely different from that of this study. Besides,
the historical trend of vanillic acid shows many higher sporadic peaks during the
Little Ice Age (LIA) and extended LIA (ELIA), which is somewhat similar to
concentration trends of 10-year bin averages of p-HBA and vanillic acid from
Svalbard ice core (Grieman et al., 2018a). These similarities could be due to a similar
source and source regions. In contrast, dissimilarity of historical records of these
compounds before and after ELIA suggests that shifting of atmospheric circulation or
different spatial pattern of biomass burning and/or that climate-driven effects are
deeply involved (Pokhrel et al., 2015). Hence these results further support a snap of
biomass burning periodic cycles of Alpine glacier in the North Pacific Rim

## 3.4 Biomass burning tracers, temperature and climate: Atmospheric consequences

There is a direct relationship between the atmospheric temperature and pressure in the NH; that is, one variable (temperature/pressure) follows the same change when it comes to increasing and decreasing mode. This mechanism drives the atmospheric air mass from one place to another in the NH. For example, the semi-permanent Siberian High and Azores High drive the air mass from those regions to Alaskan (e.g., Aleutian Low) and Icelandic (e.g., winter air mass circulation) regions in the NH (Mantua and Hare, 2002). This Siberian High-pressure system (the vertical extent is up to 3 km from the surface) is one of the principal sources of polar air mass in the NH and is a principal factor to control air pollution in the Alaskan regions. Ten-day airmass backward trajectories (Fig. 7a-f) supported the same atmospheric transport pathways to southern Alaska. The consequences of such atmospheric circulation in the Alaskan region can be directly observed with the correlations of monthly (annual and seasonal) records of global lower troposphere temperature anomalies (GLTTA) with this study (Fig. 4a-o).

These pieces of evidence are further reflected by the Pacific Decadal Oscillation (PDO), which is characterized by relatively high temperature from the west to east coasts of the North Pacific Rim (Mantuna et al., 1997; Mantuna and Hare, 2002; MacDonald and Case, 2005; Shen et al., 2006). The similar trend of levoglucosan with five points running average of this PDO cycle, except for few points (e.g., 1750, 1834, 1870, 1913, 1934 and 1966) during the whole period of 1665 to 1995, represents ecological changes and/or changes in climate-driven biomass burning activities. These years, that is, 1750, 1834, 1870, 1913, 1934 and 1966 A.D., are influenced by micro and meso scale rather than synoptic and global scale weather

conditions and/or by long spikes represented by single fire events or seasonal biomass
burning activities (Fig. 5a,b). Hence, the positive/negative phase of PDO represents
zonal and/or meridional flows and elevated/depressed transport of levoglucosan to the
eastern North Pacific Rim.

In addition, winter precipitation (i.e., snowfall) is higher than usual in the

Alaskan coast. The annual precipitation of Aurora is increasing. The positive
correlations ($R^2$ or $\tau$) of levoglucosan (except for few points, 1993, 1997, 1999 and
2005), dehydroabietic (except for, 1991 and 1998) and vanillic acids (except for 1998
and 2002 ) with winter temperature (GLTTA) are 0.55, 0.44 and 0.29, respectively,
after the Great Pacific Climate Shift (see Fig. 4-o). When the pressure decreases, the
temperature decreases, transporting air mass from higher (e.g., East Asia) to lower
pressure regions (Alaska). Similarly, we found further evidence of long-range
atmospheric transport due to a strong pressure gradient between Alaskan (e.g.,
Aleutian Low) and East Asian regions (e.g., Siberian high). For example, the
correlations ($R^2$ and $\tau$) of these three compounds (except for a few points) are all
positive with seasonal (i.e., summer, autumn, and spring ) and annual records of this
temperature (see Fig. 4a-o). In addition, the terrestrial plant derived biomarker such as
homologous serious of high molecular weight fatty acids ($C_{21:0}$ to $C_{30:0}$) showed
increasing trends after the GPCS from the same ice core. These acids are emitted to
the source regions by vaporization of leaf waxes during biomass burning processes
(Pokhrel et al., 2015). Hence, these tracers are associated with synoptic scale
radiative climate forcing (e.g., radiative lapse rate or temperature inversion) from the
surface to boundary layer. The down slope winds and drainage of wind in the Alaskan
regions may be associated with PDO and El Nino Southern Oscillation (ENSO) in
summer (MacDonald and Case, 2005; Shen et al., 2006).

The remarkable increasing trend of dehydroabietic acid (ave. 128 ng/kg-ice,

range: 6.59-555, SD ±126 and median 108.8) has occurred after the GPCS (1977-

2007 AD). We found a significant correlation (Fig. 3a) between dehydroabietic acid

(except for 2005) and vanillic acid ($\tau$=0.60, p<0.01). In contrast, we found

insignificant correlations of levoglucosan with dehydroabietic acid (0.30) (except for

1981 and 1986) and vanillic acid (0.21) (except for, 1999 and 2005) after the GPCS,

that is, 1977-2007 A.D., revealing the local source emission. For example, the

biomass burning year of Alaska is 2004, which shows three times higher

concentrations of dehydroabietic acid (309 ng/kg-ice) than levoglucosan (95.3 ng/kg-

ice), suggesting that short range atmospheric transport enhances the dehydroabietic

acid under the local weather condition of Alaska.

The historical record of $\delta$D of the same ice core is well correlated with the

PDO cycle (Tsushima et al., 2015). Levoglucosan levels of this study are also allied

with periodicity of PDO (Fig. 5a,b) due to a Aleutian Low of North Pacific Ocean,

which is atmospheric air mass convergent near the southeast coast of Alaska (e.g.,

Aleutian Low represents the positive PDO). The average annual amplitude of $\delta$D

from this ice core is 30.9% (Tsushima et al., 2015). This high amplitude of $\delta$D could

not be conserved, if 100% of snow melting were occurred in the past. The coastal

record of climate change (e.g., winter storm development) of the Gulf of Alaska is

well correlated to the GPCS (1976 A.D.) in the PDO, suggesting that $\delta$D indicates the

air temperature of the saddle of the Aurora Peak.

The higher spikes of levoglucosan are similar to those of dehydroabietic and

vanillic acids from 1660s to 1970s. The positive/negative phases of both PDO

(MacDonald and Case, 2005; Trouet et al., 2009) cover all higher/lower spikes of

levoglucosan. The corresponding phase (positive/negative) of PDO varies from year
to several years and exhibits a tendency to cover historical intervals of these
compounds lasting several decades from 1660s to 1970s. The NAO's (wNAO) phase
are remain same for several years than PDO as shown in Figure (Fig. 5b). The
periodicity of NAO phase (positive/negative) does not represent the historical trends
(higher spikes/depression) of levoglucosan, dehydroabietic and vanillic acids (Fig. 5b,
c). This NAO represents atmospheric circulation between subtropical High and polar
Low (Trouet et al., 2009). In fact, NAO significantly dominates the North Atlantic
(e.g., North America) and European winter climate variabilities rather than those of
North Asia (i.e., Eurasia/Siberia), which is spontaneously inappropriate in this study.
**4. Summary and Conclusions**
This study has been conducted to better understand temporal trends of the
forest fire signals depend on the source region and proximity to the source and types
of vegetation in the source regions of southern Alaska since the 1660s A.D. Ice core
records of dehydroabietic acid, vanillic acid  and levoglucosan showed predominant
multidecadal trends, suggesting the variations of fire regimes and the proximity to the
source, and changes in atmospheric circulation, land-use and/or ecological pattern in
the mid to high latitudes ($\geq 30°$ N) at least before and after the 1830s and after the
Great Pacific Climate Shift (GPCS). Levoglucosan showed sporadic peaks during the
1660s-1830s, and single spikes in the 1898, 1913, 1966, and 2005 A.D. These spikes
indicate a significant contribution of biomass and/or biofuel burning attributing to
their source-specific emission and atmospheric stability in Alaskan regions.
Dehydroabietic and vanillic acids showed similar historical trends with
levoglucosan before the 1830s, suggesting that hard wood and conifer trees (e.g., resin
and lignin boreal conifer trees, deciduous trees and other higher plants) and perennial
grasses ($C_3$ and $C_4$ plants) were simultaneously important as burning sources. The
gradually increasing concentration trends of dehydroabietic and vanillic acids after the
1980s onward show a strong correlation ($\tau = 0.60$, $p < 0.01$; after the GPCS; 1976),
suggesting significant changes in either burning patterns (i.e., new land-use pattern or
new ecological pattern) or atmospheric circulation over Alaska by the climate driven
forces with exhibiting similar signals of biomass burning tracers compared to
insignificant correlation of levoglucosan with these compounds.

581    The significant positive correlations ($\tau$) of these three compounds with lower

tropospheric global (annual and seasonal) temperature anomalies (GLTTA) suggest
that Alaskan snow precipitation was involved with climate-driven forces at least after
the GPCS to onwards. These tracers are allied with synoptic and global scale radiative
climate forcing (e.g., radiative atmospheric lapse rate or inversion) from the surface to
atmospheric boundary layer. The series of higher (lower) spikes of biomass burning
tracers from Aurora Peak represent the positive (negative) phase of PDO periodicity
cycles in the North Pacific Rim. The correlation of temperature and comparison with
PDO cycle with this study are further covering the excellent signal of periodic cycle
of climate-driven regime, that is, atmospheric activities, climate and weather
conditions, ecological changes, and individual fire activities of source regions to the
Aurora site.

593    The straight-forward historical trends of these three compounds were

significant before the 1830s, which differ from the Kamchatka ice core record,
suggesting that there are some differences between the western and eastern North
Pacific Rim for the emission, frequency, and deposition. The concentrations of these
three compounds from Aurora Peak are higher than those from other ice core studies
in the NH (e.g., Kamchatka, Svalbard, Tunu, and Akademii Nauk). In contrast, there
are similarities of depressed concentration trend of Aurora with other ice core studies
at least for one hundred years (e.g., 1890-1980s: Akademii Nauk, 1820-1960: Tunu
Greenland), suggesting that sources of biomass burning tracers are further confined
within the same regions, traveling from very long distances and are well captured
within the snow particles. If it is true, these compounds might be involved as cloud
condensation nuclei from the surface to 15.2 km, (i.e., cumulonimbus cloud),
transporting thousands kilometers to Aurora. It bounds positive feedback for the
climate change and/or climate variability in the North Pacific Rim.
**Acknowledgements**
This study was partly supported by the Japan Society for the Promotion of Science
(JSPS) through grand-in-aid Nos. 19340137 and 24221001 and Japan student service
organization (JASSO). We also acknowledge the support from the Institute of Low
Temperature Science, Hokkaido University for the ice core program.

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

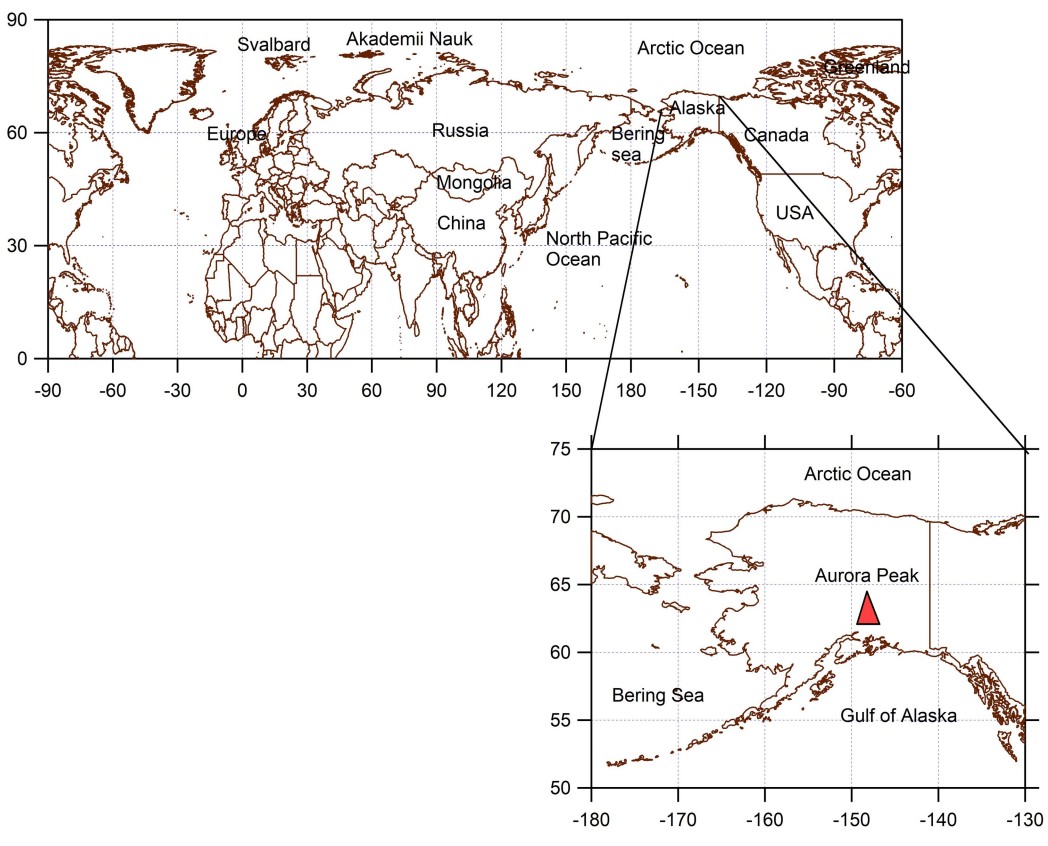


Figure 1. Geographical location of Aurora Peak in Alaska, where a 180-meter long ice core was drilled on the saddle of this peak in 2008

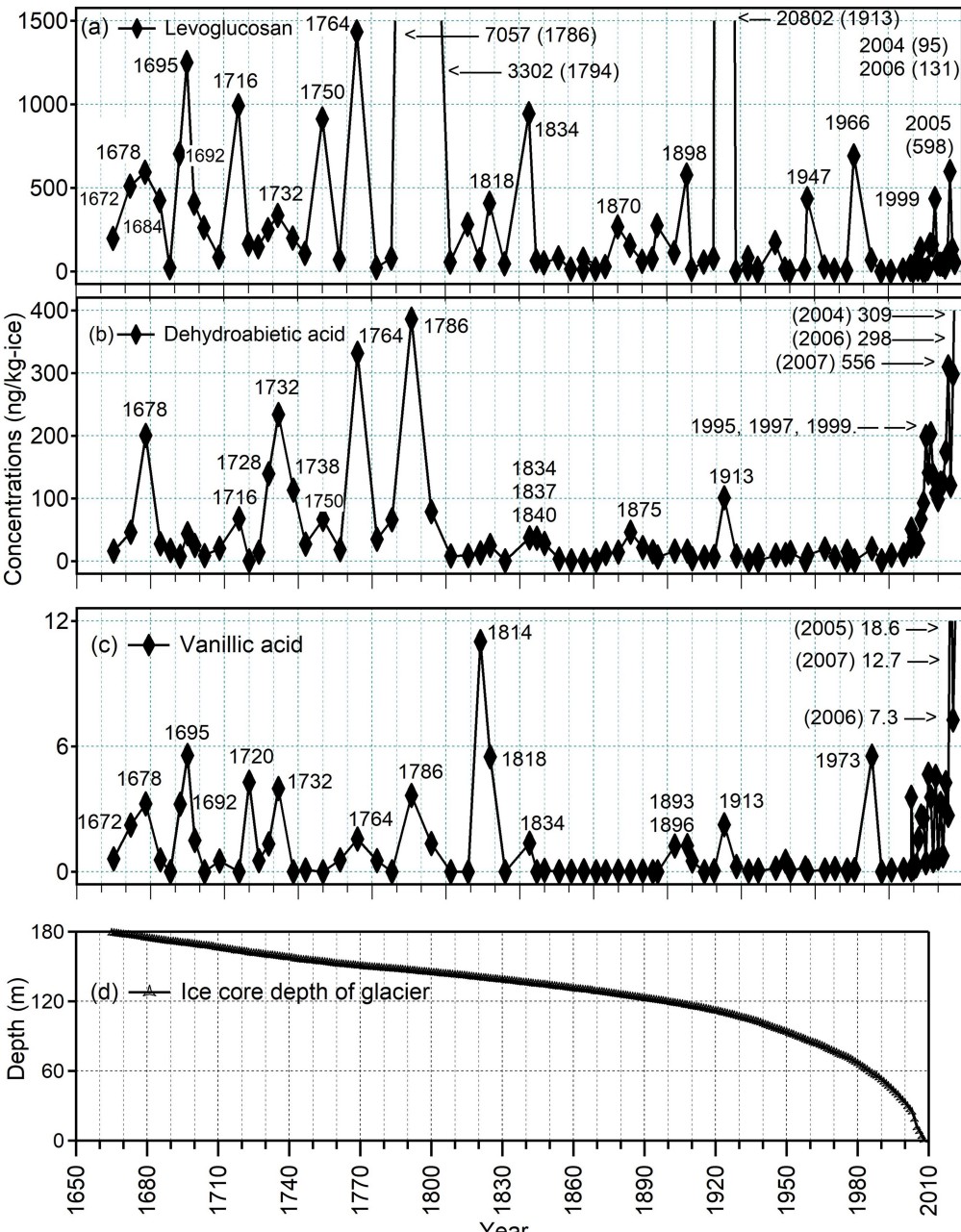


Figure 2. Concentration changes of (a) levoglucosan, (b) dehydroabietic, (c) vanillic
acids in the ice core, and (d) depth of the ice core collected from Aurora Peak in
Alaska for 1665-2008 A.D.

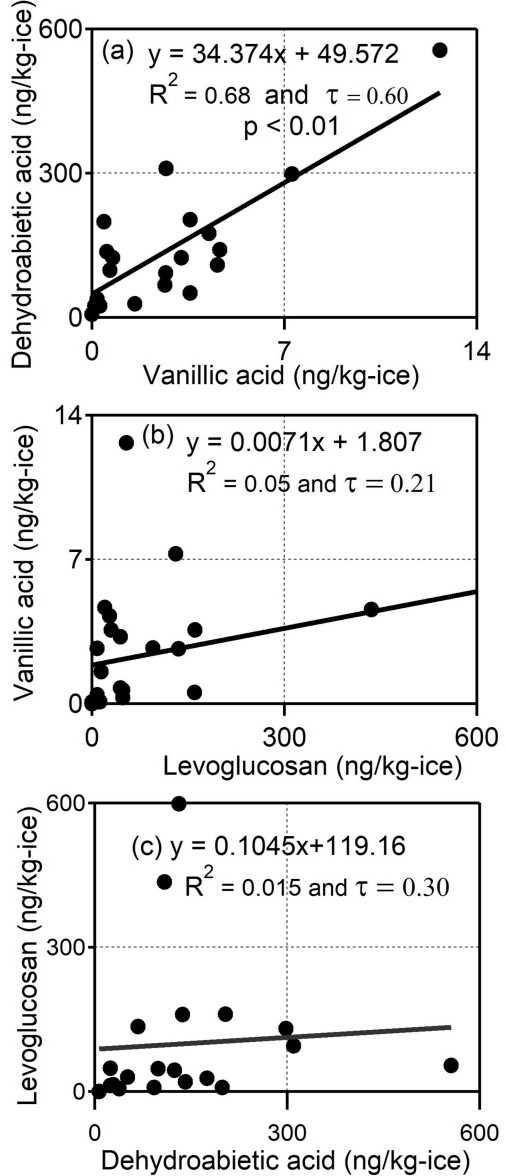


**Figure 3**. Correlations (Pearson: $R^2$ and Kendall: $\tau$) plots between the concentrations of (a) dehydroabietic and vanillic acids, (b) vanillic acid and levoglucosan, and (c) levoglucosan and dehydroabietic acid. In (b) and (c), correlations are insignificant in the Alaska ice core records from the saddle of Aurora Peak after the Great Pacific Climate Shift (1977-2007 A.D.).

880

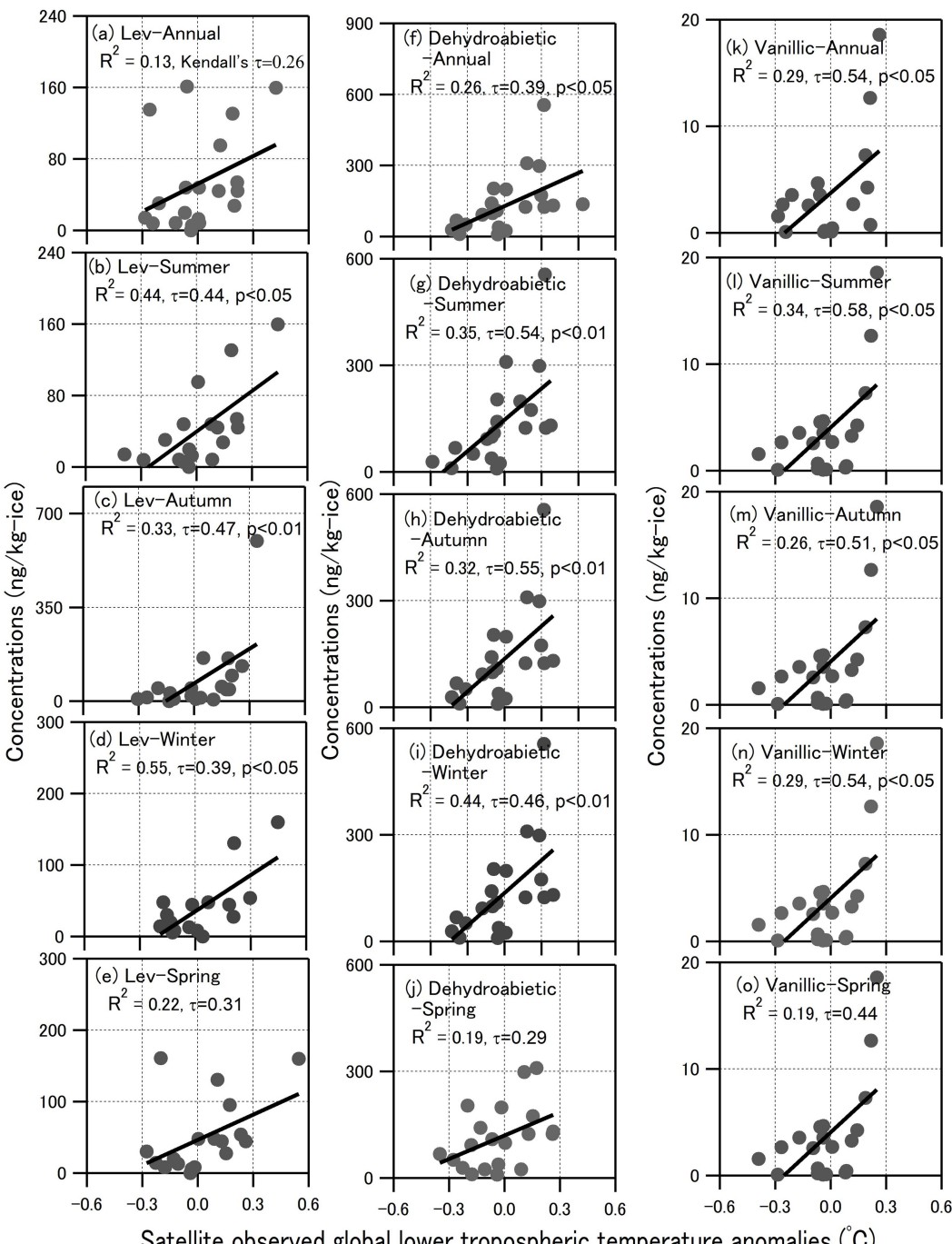

**Figure 4.** Correlation (Pearson: $R^2$ and Kendall: $\tau$) plots between satellite-observed global lower tropospheric temperature anomalies (i.e., microwave sounding unit temperature anomalies (°C) of annual and seasonal records) and annual concentrations of (a-e) levoglucosan, (f-j) dehydroabietic acid, and (k-o) vanillic acid after the Great Pacific Climate Shift in the Northern Hemisphere.

888

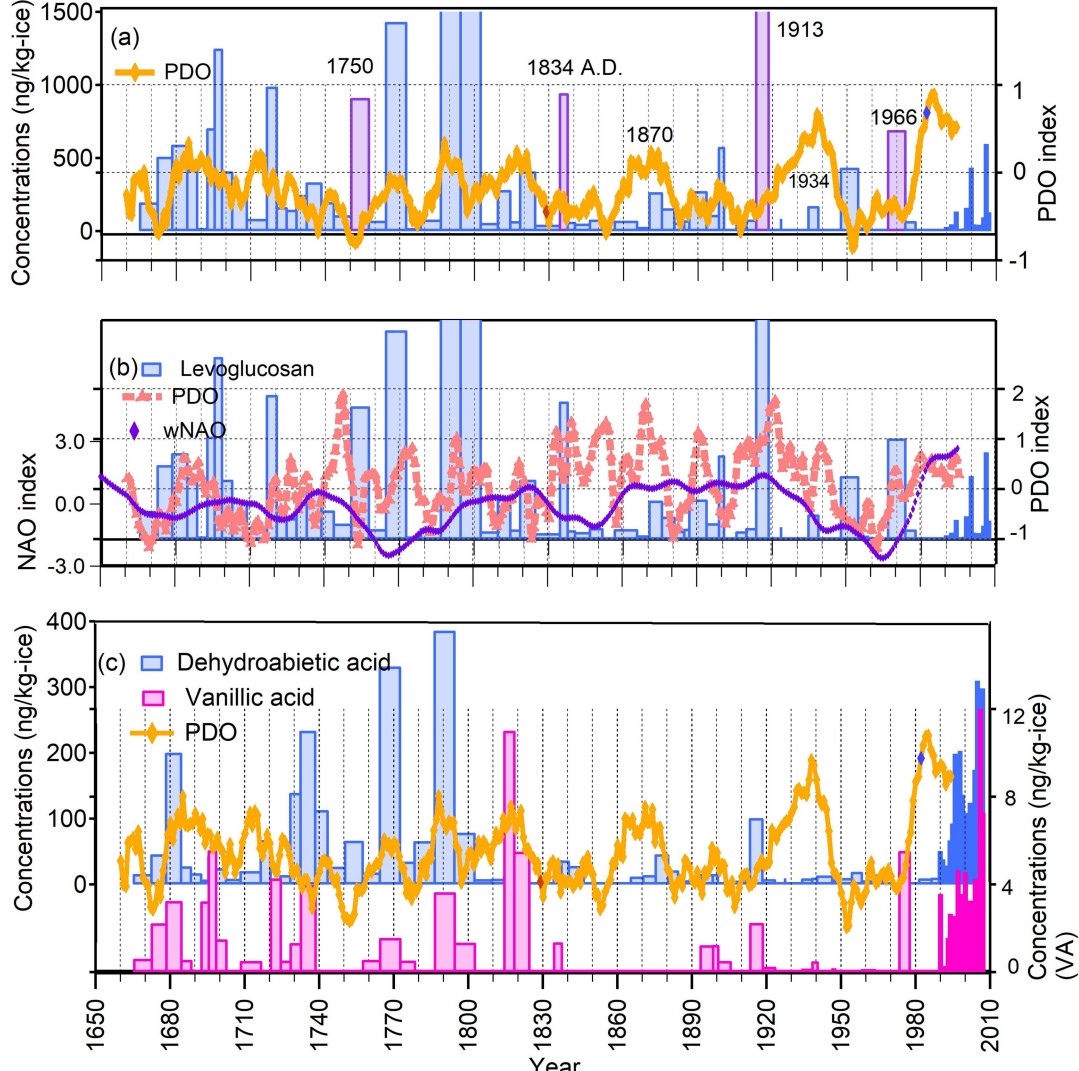

889

**Figure 5.** Historical trends in the concentrations of (a) levoglucosan (Aurora Peak) and Pacific Decadal Oscillation (5 year mean PDO) index (Shen et al., 2006), (b) levoglucosan (Aurora Peak), PDO-5 year mean index (MacDonal and Case, 2005) and Multi-decadal winter North Atlantic  index (wNAO) (Trouet et al., 2009), and (c) dehydroabietic and vanillic acids and PDO for 1665-2008 A.D.

895

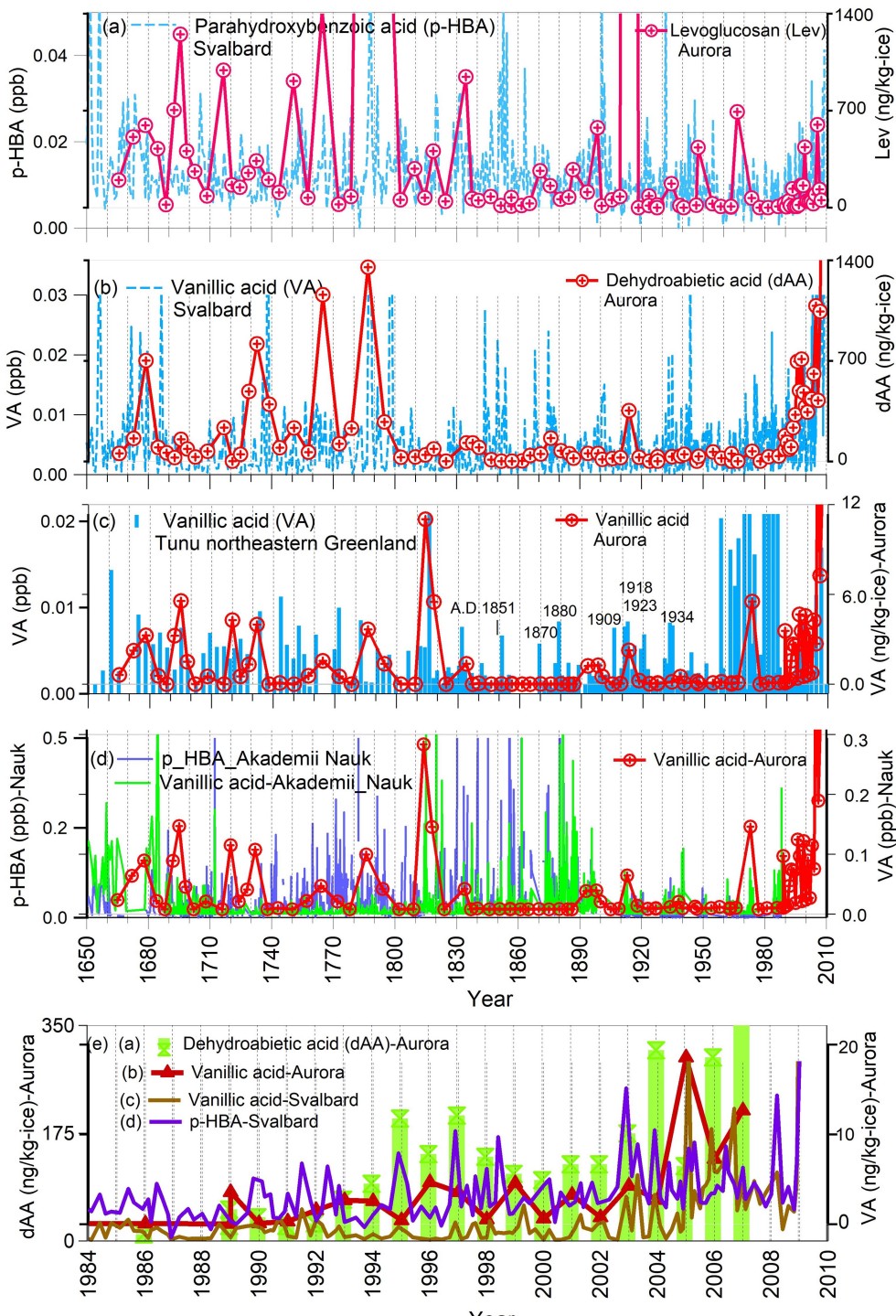

896

**Figure 6**. Historical trends of (a) p-hydroxybenzoic acid (p-HBA) of Svalbard, (b) vanillic acid (VA) of Svalbard, (c) VA of Tunu Greenland, (d) p-HBA and VA of Akademii Nauk, with levoglucosan (Lev), dehydroabietic acid (dAA) and VA of Aurora Peak, respectively, and (e) historical trends of dAA and VA of Aurora and VA and p-HBA of Svalbard after the Great Pacific Climate Shift (1977-2007 A.D.).


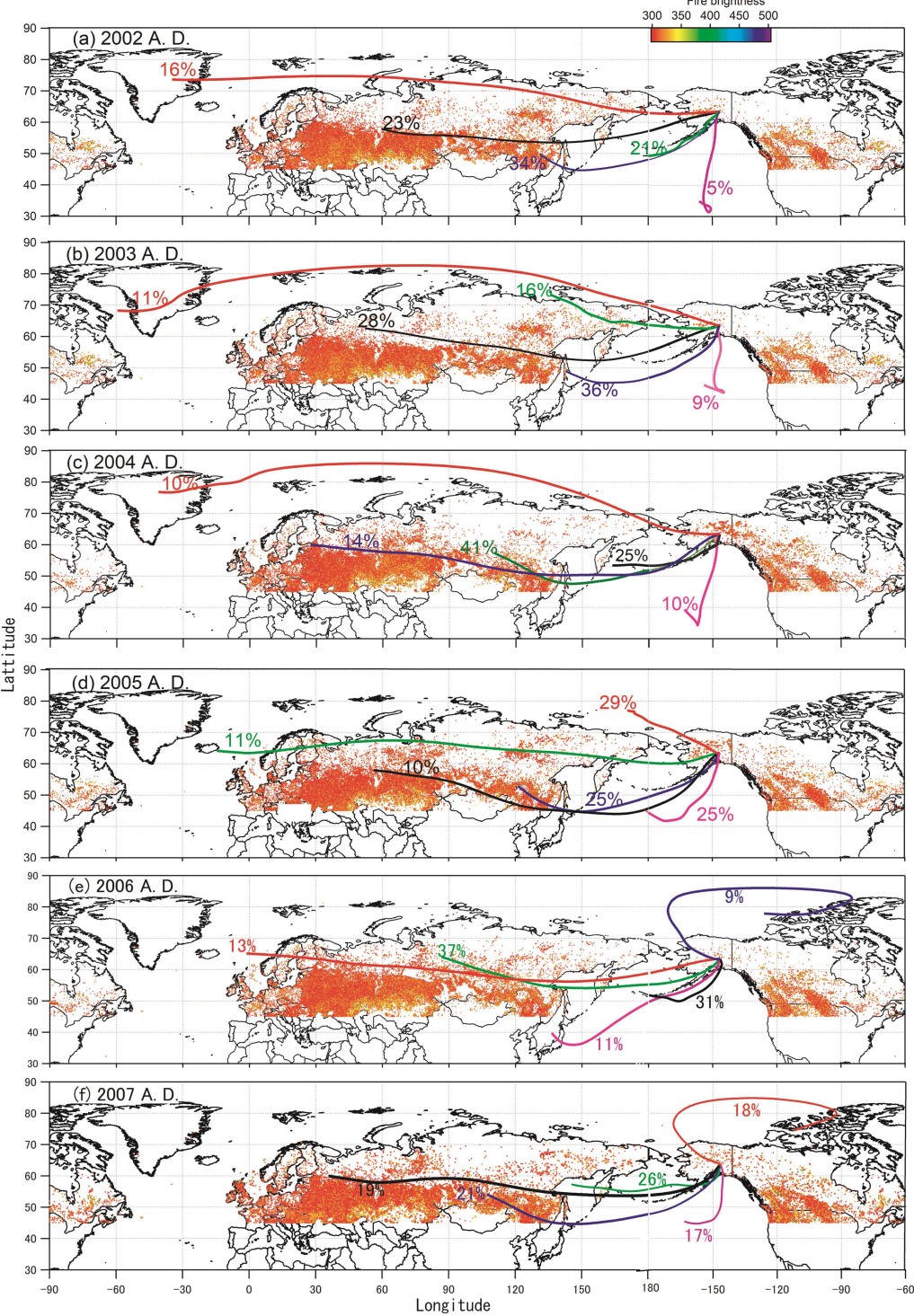


Figure 7. The MODIS fire spots together with 10 days back trajectories analysis (a-f)
of Alaskan regions since 2001 to 2007.