# Peer review of "Ice core records of levoglucosan, dehydroabietic and vanillic acids from Aurora"

_Atmospheric Chemistry and Physics, 2019_

## Referee Comment (RC1) · Anonymous Referee #1 · 27 Mar 2019

GENERAL COMMENTS: The authors present a record of three biomass burning tracers (levoglucosan, dehydroabietic and vanillic acids) from the Aurora Peak ice core from southern Alaska. The North Pacific contains few long fire records, and this study adds a valuable location for a biomass burning record. I commend the authors for determining multiple fire tracers within the same ice core and for investigating the different types of information that can be gained from each marker. The authors compare their results with the Kamchatka ice core as well as multiple Greenland ice cores. The compari-

son with Kamchatka is much more applicable, as these cores are both within the North Pacific region. However, the authors often base much of their reasoning on the comparison with Greenland ice cores, which are almost half a hemisphere away from Aurora Peak. While such a comparison can be useful, one would not expect fire histories to be similar, due in part to this long distance between the sites. Many fire peaks in the record are single spikes, which may be due to individual fires or to relatively close-by fires. Comparing single spikes between southern Alaska and Greenland is somewhat futile, unless these peaks extend over a longer time period (such as a decade or more) where the peaks can be ascribed to increased fire activity rather than individual fires. The authors do have a good data set, which adds value to both the fire science and paleoclimate communities. However, the conclusions often overreach what information the data can supply.

SPECIFIC COMMENTS:

Page 2, Line 24: Do you mean to imply that there are multiple sporadic peaks within the individual years AD 1913 and 2005? If so, then keep the sentence as is. Do you perhaps mean that AD 1913 and 2005 are individual peaks during a time period where there are few other peaks? If so, then please clarify in the abstract.

Page 2, Line 29: Where are the other ice core studies? In possible source regions? In other Northern Hemisphere locations such as the Tibetan Plateau or Greenland? Or general ice core studies of levoglucosan including in the Southern Hemisphere?

Page 3, lines 42-44 and continuing throughout the paper: Choose to list references in either chronological or alphabetical order, and then remain consistent with this decision throughout the paper.

Page 3, lines 46-52: Over what time periods and resolutions do these discrepancies exist? Decades, millennia, etc? Do you mean that the discrepancies are between the Northern and Southern Hemispheres? Are you suggesting that transport differs by hemisphere, and if so, how?

Page 3, lines 54-58: You mention that there are "a few studies" and then you cite only a single study. There are multiple studies of these biomass burning markers in the Northern Hemisphere including (but certainly not restricted to) the following studies:

Grieman, M.M., Aydin, M., Isaksson, E., Schwikowski, M., Saltzman, E.S. (2018). Aromatic acids in an Arctic ice core from Svalbard: a proxy record of biomass burning. Climate of the Past, 14, 5, 637-651, DOI: 10.5194/cp-14-637-2018

Li, Q., Wang, N.L., Barbante, C., Kang, S., Yao, P., Wan, X., Barbaro, E., Hidalgo, M.D.V., Gambaro, A., Li, C.L., Niu, H.W., Dong, Z.W., Wu, X.B. (2018) Levels and spatial distributions of levoglucosan and dissolved organic carbon in snowpits over the Tibetan Plateau glaciers. Science of the Total Environment, 612, 1340-1347, DOI: 10.1016/j.scitotenv.2017.08.267

Parvin, F., Seki, O., Fujita, K., Iizuka, Y., Matoba, S., Ando, T., Sawada, K.(2019) Assessment for paleoclimatic utility of biomass burning tracers in S-Dome ice core, Greenland. Atmospheric Environment, 196, 86-94. DOI: 10.1016/j.atmosenv.2018.10.012

You, C., Yao, T., Xu, C. (2019) Environmental Significance of levoglucosan records in a central Tibetan ice core. Science Bulletin, 64, 2, 122-127, DOI: 10.1016/j.atmosenv.2018.

Zennaro, P., Kehrwald, N., Marlon, J., Ruddiman, W.F., Brucher, T., Agostinelli, C., Dahl-Jensen, D., Zangrando, R.,Gambaro, A., Barbante, C. (2015) Europe on fire three thousand years ago: Arson or climate? Geophysical Research Letters, 42, 12, 5023-5033, DOI: 10.1002/2015GL064259

Page 3, lines 58-61: Do you mean that this is the first time that all three specific biomass burning tracers (levoglucosan, dehydroabietic and vanillic acid) were analyzed together in an ice core? Or do you mean that this is the first time that Kawamura et al. investigated these three markers?

Page 4, Line 62: Please be clear as to which three compounds you are investigating in the paper.

Page 4, Line 63 and continuing throughout the paper: Acronyms can tend to interrupt the flow of reading instead of helping. Use the name "Aurora Peak" throughout the rest of the paper instead of the acronym "APA".

Page 4, Line 64 and continuing throughout the paper: All dates must include C.E. or A.D.

Page 4, Line 71 and continuing throughout the paper: Many of your references contain both the numbers from your citation software, followed by the written names of the references. Carefully check the document and omit all numbers related to the references. (In this case, you would change "Eastern Europe2 (Kawamura et al., 2012)" to "Eastern Europe (Kawamura et al., 2012)".

Page 4, Line 80: Here you cite Tsushima, 2014 and Tsushima et al., 2014. Your references state that both were published in 2015. Please double-check these dates.

Page 4, Line 86: Pokhrel et al., 2015b does not exist in your reference list. Instead, you cite a paper that was submitted to ES&T in 2016. In a search for this paper on "Web of Science" this paper was not published. It is not acceptable to use unpublished – and possibly rejected – work as a reference.

Page 4, Lines 62-72: This paragraph ends abruptly. The authors state the aim of the study at the beginning of the paragraph, then dive into specifics, and then abruptly stop. The study goal becomes lost in this paragraph structure. Please revise.

Page 4, Lines 74-81: What evidence do you have that the ice core is 274 y BP at the depth of 180 m other than the annual layer counting? Cite Tsushima, 2015 a and b here. (Your citation of Tsushima, 2014 is incorrect as the papers you cite were published in 2015. Also, one of the Tsushima papers should be labeled "a" and the other should be "b"). In the Pokhrel, 2015 dissertation abstract, the age at 180 m

is cited as 343 y BP. Of course, with new knowledge, depth-age scales can change. However, what DID change? Did you acquire independent dates?

Page 4, Lines 80-81: Do you mean that 5-10 mm was shaved off the outside section of the core? Off all sections? How did you clean the ceramic knife?

Page 4, Line 84: What kind of container? LDPE? Glass? What size? Were these containers cleaned? If so, how?

Page 4, Line 85: What do you mean by a "standard clean room"? A Class 100? A Class 10,000?

Page 4, Lines 87-88: Do you mean one quarter of the ice core by depth or by circumference? You mention that the sampling frequency was approximately 40% of the entire ice core. However, if you are taking 25% of the core, then how do you get a sampling frequency of 40%? If you are sampling by circumference, then you would have a continuous record. If you are sampling by depth, then you would have approximately 50% of the core. Are there locations in the core that you were not able to sample due to breakage, etc? If so, then note these locations.

Page 4, Line 102: What are the "authentic standards" that you used? What concentrations? Where did you purchase the standards? Did you use any isotopically-marked standards? Did you include the standards in the samples as internal standards?

Methods and materials: Do you have any lab blanks? Do you have any procedural blanks? If so, do you have any way for accounting for the concentrations of the three analytes in your blanks? If you do not have any blanks, what sort of QA/QC measures did you apply? What is the LOD for the analyzed compounds? Was each sample analyzed in triplicate (Page 5, Line 105)? Or were only three samples analyzed in triplicate?

Page 4, Line 104: You mention that analytical details are included in Simoneit et al., 2004, yet you do not include this paper in your references. A literature search demonstrates two options where Simoneit et al., 2004 investigates levoglucosan, etc. However, neither of the two possible Simoneit et al., 2004 papers nor Fu et al., 2008 (who you mention as another paper that contains the full method details) include essential information such as the m/z, the amount of time analyzing each m/z, etc. While you can still cite these papers, you still do need to include the primary information of the method.

Page 6, Line 116: When you state an "important fraction", please mention how this fraction is important. Due to the volume? Due to the fact that anhydromonosccharides were produced by biomass burning?

Page 6, Lines 125-128: Here you argue that Figure 1 demonstrates that Aurora Peak is far away from any biomass burning sources, but in Figure 5 you demonstrate that there are fire sources near the peak. Explain this discrepancy.

Page 6, Line 119: Here, you state that levoglucosan is only produced at temperatures above 300ïĆř C, and cite references from 1984 until 2002. However, more recent literature (Kuo et al., 2011) states that levoglucosan and its isomers are only produced at temperatures up to 350ïĆř C. Explain this discrepancy. The citation for Kuo et al. is the same as you use later on the same page:

Kuo, L-J., Louchouarn, P., Herbert, B.E. (2011) Influence of combustion conditions on yields of solvent-extractable anhydrosugars and lignin phenols in chars: Implications for characterizations of biomass combustion residues. Chemosphere, 85, 797-805 doi:10.1016/j.chemosphere.2011.06.074

Section 3.1: What evidence do you have that the fact that mannosan and galactosan are consistently below the limit of detection is due to these isomers not being present, versus to these isomers simply not being detectable by the analytical method? As mannosan is below the detection limit, the statement that "Thus, levoglucosan/mannosan mass ratios (L/M) could be relatively high" does not make sense. You would have to divide your levoglucosan results by zero. As you do not have quantifiable numbers

for mannosan and glacactosan, the paragraph (Page 7, Lines 143-151) does not add value to the paper, and can be omitted.

Page 8, Lines 171-174: Many studies have wide ranges for the atmospheric lifetime of levoglucosan. The Hennigan et al., 2010 study is on the extremely low end of these calculations. Please include other studies and results to give a more accurate range.

Page 8, Line 175 to Page 9, Line 189: Do you mean that the source regions are southern Alaska as well as the possible source regions that are listed in lines 181-182? If so, then why do you separate southern Alaska? Do you mean that the heavy forest fires in eastern Siberia occur now, or occurred in the past, or both? Why do you compare your regional record to global biomass burning? Do you have a regional biomass burning record from charcoal or other data? Do you have any indication of land-use change in eastern Siberia in the 1840s?

Page 9, Lines 190-200: This paragraph is illogical. Please omit the sentence "We did not detect significant concentrations of any isomers as we have discussed above" as discussing isomer ratios does not fit into this paragraph.

Section 3.1: Are these spikes individual points or are they multiple points consecutively in the ice core? How can you determine if the spikes indicate a close fire, versus a fire that is farther away? Do the spikes indicate fire intensity?

Page 9, Lines 201 to 204: The argument "These suggest that ice core $NH_4+$ has common sources in the circumpolar regions" does not logically follow from the preceding sentence.

Pages 9 and 10: The distances between Alaska and Greenland is thousands of kilometers. Do you have other evidence than similar spikes of $NH_4+$ between Mt Logan, GISP 2, and 20D to suggest "that ice core $HN_4+$ has common sources in the circumpolar regions?" Are these spikes just visually the same, or is there some statistical test done to determine that these spikes correlate? Are the spikes just individual points,

or are they peaks over decades? The following review paper investigates boreal fire source regions and the atmospheric transport, with implications for your assumptions on pages 9 and 10. Essentially, it would be difficult, although not impossible for the fire source regions to be the same for both Alaskan and Greenland ice core records:

Legrand, M., McConnell, J., Fischer, H., Wolff, E.W., Preunkert, S., Arienzo, M., Chellman, N., Leuenberger, D., Maselli, O., Place, P., Sigl, M., Schuepbach, Flannigan, M. (2016) Boreal fire records in Northern Hemisphere ice cores: a review. Climate of the Past, 12, 2033-2059, doi:10.5194/cp-12-2033-2016

Section 3.1: Why is Whitlow et al., 1994 your primary reference when the research into boreal forest fires and ice core records has increased substantially in the past 25 years?

Pages 10 and 11: Lines 232-239: Would you like to say that the fires after 1900 only affect Mt. Logan and not Aurora Peak? Why is Aurora Peak more similar to Greenland records than to other Alaskan records? Please clarify.

Page 11, Lines 245-246: What mechanism do you propose for increased dilution and/or scavenging of biomass plumes after the 1830s? Would this mechanism affect the entire Arctic or just southern Alaska?

Page 11, Lines 245-247: In what ice core(s) do the see the difference in the concentrations? Is this a comparison between Aurora Peak and Greenland ice cores? If so, why would you expect a similarity over half a hemisphere of distance?

Page 11: Lines 247-249: You state "These special events further suggest that Alaskan glaciers cannot preserve most biomass burning events in the circumpolar regions, which occurred in the source regions of Siberia and North America". Do you mean that the combination of distance and atmospheric transport means that most fires in Siberia and North America will not be recorded in Alaska mountain glaciers? Do you mean that the difference between the Alaskan and Greenland ice core biomass burning records

suggest that the Alaskan glaciers are not as good of receptors as the Greenland ice sheet?

Page 11, Lines 257-258: By mentioning the Little Ice Age, do you mean to imply that the cooler weather influenced the decreased biomass burning? Do you have any evidence or records of decreased temperatures in Siberia or North America during this time period?

Page 13, line 287-289: Do you mean that the East Asian regions are more important for regional levoglucosan production or that they are more important as a source of levoglucosan for Aurora peak? You do go on to discuss this point further in the next few paragraphs, but as this is the first time that the reader is exposed to this idea, it is better to be clear from the onset.

Pages 14 and 15: Why would the long-range atmospheric transport be insignificant? It may just be that regional transport overwhelms the long-range signal from the 1920s until the present, but the same amount of long-range transport may occur. What do you mean that the concentrations "are secure"?

Pages 14 and 15: Why did you choose to do a point to point correlation of data from a time series? The data are definitely skewed, with the majority of the data with low concentrations and then a few separate spikes. Why did you not choose other types of correlations that may be more applicable? What is the statistical level of correlation of each of these factors?

Page 17, Line 400: In what way do the forest fire signal depend on the source region? Do they depend on proximity to the source? Do they depend on the type of vegetation burned in the source region?

Page 17, Lines 401-403: Your paragraph is better without this sentence. Please remove.

Figure 5: Why do you investigate the MODIS fire spots over different areas for each

year? For example, the 2004 plot stops at ∼45 degrees N, while the 2006 plot stops at ∼ 32 degrees N. Why do you include much of the United States, if regions south of 45 degrees north are unlikely to be a source region? If Siberia is a major source region, why did you not also investigate Siberia with any available data?

TECHNICAL CORRECTIONS:

Title: Place "the" before "1660s" in the title

Page 2, Line 18: Change "melt" to "melted"

Page 2, Lines 24 and 25: Change "there are few discrepancies of higher spikes among them after the 1970s" to "there are a few discrepancies of higher spikes especially after the 1970s".

Page 1, Line 27: Place "as well as" before "other higher plants"

Page 2, Line 29: Change "regions of southern Alaska, being different from previous ice core studies" to "regions of southern Alaska. These results differ from previous ice core studies."

Page 2, Line 34 and Page 17, Line 398: The word "gleaming" is wonderful, and I am sorry to suggest replacing this word with more boring options. Unfortunately, it is not quite clear what you would like to suggest with this word. Do you mean substantial? Clear? Definitive? If so, please use one of these words.

Page 3, Line 39: Replace "provide the" with "archive"

Page 3, Line 42: Omit "which are reported elsewhere"

Page 3, Lines 45-46: with "have some extent on climate change effect" do you mean to say "may affect climate change"? If so, then replace the phrase.

Page 3, Line 54: Place "a" before "pyrolysis"

Page 4, lines 66-67: Change "Particularly, 10 day backward trajectory" to "Particularly,

10-day back trajectories"

Page 4, Lines 85-86: By "All steps are followed as reported previously prior to analysis (Pokhrel, 2015; Pokhrel et al., 2015b)" do you mean "All analytical steps are previously reported in Pokhrel, 2015 and Pokhrel et al., 2015b"?

Page 5, Line 89: Change "melt" to "melted"

Page 5, Line 90: Change "shape" to "shaped"

Page 5, Line 93: Choose to use either chemical formulas or names, and then be consistent throughout the paper.

Page 5, Line 94: Change "reported previously" to "previously reported"

Page 5, Line 107: Change "were" to "was"

Page 5, Line 108: Change "were" to "was"

Page 5, Line 108: Omit "traject" before "compounds"

Page 5, Line 109: I am sorry, but I do not understand what you would like to say by "physical functioning fire smoldering spot". Please change this phrase.

Page 6, Line 116: Place "to" before "an important fraction".

Page 6, Line 125: Omit "(i.e., distribution)"

Page 6, Line 128: Omit "the" before "Aurora Peak" and replace "the" with "any" before "biomass burning"

Page 7, Line 144: Change "sifnificant" to "significant"

Page 7, Line 160: This is the first time that you have used the acronym "BB". Replace with "biomass burning" and do not use the acronym.

Page 8, Line 162: Replace "didn't" with "did not".

[Figure]

Page 8, Line 175: Replace "around" with "the"

Page 9, Line 192: Use the full word "levoglucosan" rather than "Lev".

Page 8, Line 183 and Page 9, Line 198: Replace "borel" with "boreal"

Page 9, Line 204: Replace "got" with "obtained"

Page 10, Line 212: Change "is consistent to" to "is consistent with"

Page 11, Line 238: Replace "only the exception" with "the only exception"

Page 11, Line 240: Replace "Above results and discussion suggest the subsequent evidences" with "The above results suggest"

Page 11, Line 242: Replace "souhtern" with "southern"

Page 11, Line 251: Unfortunately, "heavy" does not describe forest fires. Do you mean intense? Do you mean widespread?

Page 11, Line 249: Replace "Siberian" with "Siberia"

Page 11: Line 248: Replace "can not" with "cannot"

Page 11, Line 254: Replace "declined" with "declining"

Page 12, Lines 280-283: Change to "These results suggest that biomass burning plumes of pine, larch, spruce and fir trees in Siberian regions (Kawamura et al., 2012: Ivanova et al., 2010) have a substantially larger influence on Kamchatka, southeastern Russia than on southern Alaska".

Page 12, Line 286: Change "borel" to "boreal"

Page 13, Line 296: Change "discrepancy" to "discrepancies"

Page 13, Lines 297 to 298: Change "Kamchatka showed gradual increase after the 1950s" to "Dehydroabietic acid concentrations gradually increased in the Kamchatka ice core after the 1950s".

Page 13, Line 305: Change "doesn't" to "does not".

Page 14, Line 326: Change "conifer rich" to "conifer-rich"

Page 15, Line 347: Remove "from" before "climate driven"

Page 15, Line 349: Change "could be" to "may be"

Page 16, Line 369: Replace "constitute" with "constituent"

Page 16, Lines 377-384: You only use the acronym "NPR" three times. It is much better to use the words "North Pacific Rim" than an acronym that introduces confusion. Please use the words for this phrase rather than the acronym throughout the paragraph.

Page 17, Line 395: Replace "with early study" with "than a previous study" and then cite the study in the sentence.

Figure 1: Place "a" before "180-meter". This figure does not need a citation unless you are using the exact figure as in your earlier work.

[Figure]

---

## Referee Comment (RC2) · Anonymous Referee #2 · 10 Apr 2019

"Ice core records of biomass burning tracers (levoglucosan, dehydroabietic and vanillic acids) from Aurora Peak in Alaska since 1660s: A new dimension of forest fire activities in the Northern Hemisphere" by Ambarish Pokhrel and co-workers.

**Overall evaluation:**

The topics is of importance since fires are a major source of gases and aerosols that strongly impact chemical composition of the atmosphere and the radiation balance. In turn, climate changes directly disturb the fire regime, for instance through the duration of fire weather conditions and changes in vegetation, particularly in the boreal regions. In addition to this overall interest, the data presented in this paper would be useful to discuss the consistency between these three organic markers (levoglucosan, dehydroabietic and vanillic acids) and other potential proxy including ammonium. However, as it stands, the paper suffers from too many weaknesses to be recommended for publication at the ACP journal. I recommend to the authors to take time to revisit the existing literature and their data.

**Major weaknesses:**

1. Quality of the Aurora ice record: Some key information are missed in the manuscript for the reader to evaluate the quality Aurora ice core record. Indeed, when using an ice core record to infer atmospheric information, the reader (and the reviewers) needs to have some basic information that are not given in section 2 (Materials and Methods). I think that, given the rather low elevation (2850 m) of the Aurora site, we may expect frequent melting. If, so that has to be clearly stated in the manuscript and the authors would discuss the possible consequence for the quality of the ice record in terms of atmospheric signal. Since the effect of melting is not well know for organics, it would be nice to show the record of major ions (including ammonium, nitrate, and sulfate). Checking your Figure 4, i am very surprised by the nitrate levels that are shown to range between 0 and 34 ppb (i.e., very low levels). At the opposite, the nss-K level (ranging from 0 to around 50 ppb) exhibits several values exceeding 15 ppb (which is a lot): how much abundant is calcium in this core? (see my further comments on the use of fine potassium).

2. Inconsistencies: Since you will discuss in section 3 (as also mentioned in the abstract) the correlations of levoglucosan with $NO_2^-$, $NO_3^-$, nss-$SO_4^{2-}$, nss-$K^+$, and $NH_4^+$ that are all insignificant (suggesting that these anions and cations do not represent a gleaming signal of biomass burning activities in the source regions for southern Alaska)), it would be nice to show the profiles. This need to report these profiles also comes from the fact this observed absence of correlations contrasts with the statement that I find in the paper from Tsushima et al. (2005) stating "To confirm the dating based on D and $Na^+$ seasonal cycles, we compared the dating of the ice core with reference horizons of known age (Fig. 3). We found a large peak of $NO_3$ and $NH_4^+$ and a visible dirty layer at 8.55 m w.eq., which we ascribed to the year 2004. Generally, $NO_3$ and $NH_4^+$ are released by forest fires (e.g., Legrand and Mayewski, 1997; Eichler et al., 2011). »

This point clearly needs to be discussed showing all the records, i think. Since, as also suggested by your figure 5, the 2004 year was characterized by large fires in Alaska, please also comment your organic records for this year ???

3. Numerous previous works are not cited or adequately cited in the manuscript:

In the introduction and at several places in the text, the previous works done on fire records in ice cores are not adequately cited, and some important references are missed including two reviews papers (see the list below). For example, you extensively cited the paper from Whitlow et al. (1994) for ammonium and nitrate biomass burning events that just follows the pioneering study from Legrand et al. (1992) for ammonium, nitrate and carboxylates. After the publication of these two papers, it becomes clear that, although some ammonium spikes are sometimes accompanied by nitrate peaks, it is not a general rule (Savarino and Legrand, 1998). This point was extensively discussed in the review from Legrand et al. (2016). The same is true for the non-sea-salt and non-dust potassium fraction. On this topic, in your manuscript I would recommend to report nss-non-dust-potassium (calculated by using your calcium data).

Finally, none of the Greenland ice core studies reported a sulfate perturbation with biomass burning peaks. So I will be more careful about that at line 246.

Legrand M., M. De Angelis, T. Staffelbach, A. Neftel, and B. Stauffer, Large perturbations of ammonium and organic acids content in the Summit Greenland ice core, fingerprint from forest fires ?, *Geophys. Res. Lett.*, 19, 473-475, 1992.

Legrand M., and M. De Angelis, Light carboxylic acids in Greenland ice: A record of past forest fires and vegetation emissions from the boreal zone, *J. Geophys. Res.*, 101, 4129-4145, 1996.

Savarino, J., and M. Legrand, High northern latitude forest fires and vegetation emissions over the last millenium inferred from the chemistry of a central Greenland ice core, *J. Geophys. Res.*, 103, 8267-8279, 1998.

Legrand, M., McConnell, J., Fischer, H., Wolff, E. W., Preunkert, S., Arienzo, M., Chellman, N., Leuenberger, D., Maselli, D., Place, P., Sigl, M., Schüpbach, S., and Flannigan, M.: Boreal fire records in Northern Hemisphere ice cores: A review, *Clim. Past*, 12, 2033-2059, doi:10.5194/cp-12-2033-2016, 2016.

Rubino, M., D'Onofrio, A., Seki, O., and Bendle, J.A., Ice-core records of biomass burning, The Anthropocene Review, vol. 3(2), 140-162, DOI : 10.1177/2053019615605117, 2016.

Grieman, M. M., Aydin, M., Isaksson, E., Schwikowski, M., and Saltzman, E. S.: Aromatic acids in an Arctic ice core from Svalbard: a proxy record of biomass burning, Clim. Past, 14, 637-651, https://doi.org/10.5194/cp-14-637-2018, 2018.

4. The wording is sometimes too vague or unclear, figures are unclear:

I don't understand the meaning of "A new dimension of forest fire activities in the Northern Hemisphere" in the title.

Line 29: what do you mean with "different ice core studies": Please specify from where ?, with which proxy ?, which time period ???

Lines 236 and 239: what are these numbers: concentrations of ammonium ? of levoglucosan ?

Why figure 3 reports levoglucosan only up to 600 ng L$^{-1}$ and Figure 4 only up to 1200 ng L$^{-1}$ while Figure 2 indicates levo as high as 20802 ng L$^{-1}$ ???

Please show satellite data for Siberia as well in Figure 5.

**5. Information derived from Back-ward trajectories:**

I think you may address more details on the origin of air mass reaching the Aurora site. In the present manuscript you stated at the end of the introduction: "Particularly, 10 day backward trajectory from 1992-2002 showed that southern Alaska can receive air masses from the North Pacific Regions, East Asia, Eastern Russia-Siberia, higher latitudes of Alaskan regions, Japan, and Canadian regions in the troposphere (>300 hPa) (Yasunari and Yamazaki, 2009)."

I here recommend to address the following points (that would need new calculations): (1) focus on the fire season (from June to August), (2) I think 300 hPa is too high (it is the upper troposphere) and 500 hPa (around 5 km elevation) is likely more relevant for the travel of plumes. Also check the sensitivity between 5 and 10 days.

**6. Discussion with previous records (section 3.4):**

Why do you extensively discuss your organics with ammonium records from 20D (Greenland) ???? This discussion is not very useful since the records were obtained with different proxy and are expected to be influenced by different source regions (Canada for Greenland versus Alaska and may be Siberia for Aurora).

Instead, again please show your own (Aurora) data on ammonium, nitrate, potassium, sulfate etc.

End of the review.

---

## Referee Comment (RC3) · Anonymous Referee #3 · 10 Apr 2019

This manuscript presents records of the biomass burning tracers levoglucosan, dehydroabietic acid and vanillic acid from an ice core retrieved at Aurora Peak, Alaska, and covering the time period from ca. 1660 to 2009. In general, this seems to be a high-quality data set, which may be interesting and may deserve publication, since only few ice core records of such tracers are available up to now.

Unfortunately, the manuscript does not meet basic scientific criteria, as outlined below,

[Figure]

is very descriptive and not well-written (requires English editing), and lacks a clear structure, which makes it hard to digest. Before becoming publishable, major revisions are therefore required.

- Method description is incomplete (no detection limits) and basic ice core data are missing (dating, dating uncertainty, melt extend, etc.).

- The record presented is incomplete, only 40% of the core was analysed, i.e. the records are not suitable to discuss short-term biomass burning events. Records should not be shown as continuous line; data points should indicate for which time period they are representative.

- The Aurora ice core is affected by melting with melt feature percentages of up to 100%. It should be discussed how this effects the records of biomass burning tracers.

-Recent other publications in this field should be discussed and cited (list at the end of this review).

- There is no discussion with respect to other available data. Concentrations in the Aurora core seem to be higher than in the Kamchatka core, although Aurora is located much further away from the sources.

- What is the reasoning behind conducting a correlation analysis with nss-sulphate and nss-calcium. They are not expected to have a biomass burning source. Where the ion records averaged to match the incomplete sampling of the organic tracers?

-It is unclear what can be learned from the fire spot data. Here you need to come up with a quantitative number to compare with ice core records.

- There are no conclusions, just a summary.

- Fig. 1: Already shown in Pokhrel et al., 2015 and Pokhrel et al., 2016, is this not a copyright issue?

Gambaro, A., et al. (2008). "Direct Determination of Levoglucosan at the

Picogram per Milliliter Level in Antarctic Ice by High-Performance Liquid Chromatography/Electrospray Ionization Triple Quadrupole Mass Spectrometry." Analytical Chemistry 80(5): 1649-1655.

Grieman, M. M., et al. (2017). "Aromatic acids in a Eurasian Arctic ice core: a 2600-year proxy record of biomass burning." Clim. Past 13(4): 395-410.

Grieman, M. M., et al. (2018). "Aromatic acids in an Arctic ice core from Svalbard: a proxy record of biomass burning." Clim. Past 14(5): 637-651.

Grieman, M. M., et al. (2018). "Burning-derived vanillic acid in an Arctic ice core from Tunu, northeastern Greenland." Clim. Past 14(11): 1625-1637.

Grieman, M. M., et al. (2015). "A method for analysis of vanillic acid in polar ice cores." Clim. Past 11(2): 227-232.

Kehrwald, N., et al. (2012). "Levoglucosan as a specific marker of fire events in Greenland snow." Tellus B: Chemical and Physical Meteorology 64(1): 18196.

Zennaro, P., et al. (2014). "Fire in ice: two millennia of boreal forest fire history from the Greenland NEEM ice core." Clim. Past 10(5): 1905-1924.
* * *

---

## Author Comment (AC1) · 12 Sep 2019

Responses to reviewers' comments (responses are in blue)

Anonymous Referee (ACP 2019-139) #1

Interactive comment on "Ice core records of biomass burning tracers (levoglucosan, dehydroabietic and vanillic acids) from Aurora Peak in Alaska since 1660s: A new

dimension of forest fire activities in the Northern Hemisphere" by Ambarish Pokhrel et al.

GENERAL COMMENTS: The authors present a record of three biomass burning tracers (levoglucosan, dehydroabietic and vanillic acids) from the Aurora Peak ice core from southern Alaska. The North Pacific contains few long fire records, and this study adds a valuable location for a biomass burning record. I commend the authors for determining multiple fire tracers within the same ice core and for investigating the different types of information that can be gained from each marker. The authors compare their results with the Kamchatka ice core as well as multiple Greenland ice cores. The comparison with Kamchatka is much more applicable, as these cores are both within the North Pacific region. However, the authors often base much of their reasoning on the comparison with Greenland ice cores, which are almost half a hemisphere away from Aurora Peak. While such a comparison can be useful, one would not expect fire histories to be similar, due in part to this long distance between the sites. Many fire peaks in the record are single spikes, which may be due to individual fires or to relatively close-by fires. Comparing single spikes between southern Alaska and Greenland is somewhat futile, unless these peaks extend over a longer time period (such as a decade or more) where the peaks can be ascribed to increased fire activity rather than individual fires. The authors do have a good data set, which adds value to both the fire science and paleoclimate communities. However, the conclusions often overreach what information the data can supply.

SPECIFIC COMMENTS: Page 2, Line 24: Do you mean to imply that there are multiple sporadic peaks within the individual years AD 1913 and 2005? If so, then keep the sentence as is. Do you perhaps mean that AD 1913 and 2005 are individual peaks during a time period where there are few other peaks? If so, then please clarify in the abstract. Response: Thank you for the comment. There are no sporadic peaks within the individual year. By taking the comment, we have corrected the line. Please see line 19 in the revised manuscript (MS).

Page 2, Line 29: Where are the other ice core studies? In possible source regions? In other Northern Hemisphere locations such as the Tibetan Plateau or Greenland? Or general ice core studies of levoglucosan including in the Southern Hemisphere? Response: Thank you for the comment. Other ice core studies are Svalbard, Akademii Nauk and Tunu. We have added these ice core studies in the revised MS. Please see lines 34-36.

Page 3, lines 42-44 and continuing throughout the paper: Choose to list references in either chronological or alphabetical order, and then remain consistent with this decision throughout the paper. Response: Thank you for the suggestion. By taking the suggestion, we have cited according to chronological order throughout the MS. Please see the revised MS.

Page 3, lines 46-52: Over what time periods and resolutions do these discrepancies exist? Decades, millennia, etc? Do you mean that the discrepancies are between the Northern and Southern Hemispheres? Are you suggesting that transport differs by hemisphere, and if so, how? Response: Yes. Millennia, centennials, decadal, and shorter time showed discrepancies. Hence, we have added the following lines in the text. The centennial and/or shorter time scale of trends exhibited different elevated/suppressed concentration trends of p-HBA/vanillic acid during 1600 A.D. and vanillic/p-HBA during 2000-2008 (Grieman et al., 2018a). Similarly, Svalbard ice core record (Grieman et al., 2018a) showed different elevated/suppressed historical trends/peaks with NEEM- ice core of Greenland (Zennaro et al., 2018). These results most likely suggest the changing/shifting contributions of source regions with the different ecosystem of trees, shrubs, and grasses. Please see lines 71-80 in the revised MS.

Page 3, lines 54-58: You mention that there are "a few studies" and then you cite only a single study. There are multiple studies of these biomass burning markers in the Northern Hemisphere including (but certainly not restricted to) the following studies:

Aromatic acids in an Arctic ice core from Svalbard: a proxy record of biomass burning. Grieman, M.M., Aydin, M., Isaksson, E., Schwikowski, M., Saltzman, E.S. (2018). Climate of the Past, 14, 5, 637-651, DOI: 10.5194/cp-14-637-2018 Li, Q., Wang, N.L., Barbante, C., Kang, S., Yao, P., Wan, X., Barbaro, E., Hidalgo, M.D.V., Gambaro, A., Li, C.L., Niu, H.W., Dong, Z.W., Wu, X.B. (2018) Levels and spatial distributions of levoglucosan and dissolved organic carbon in snowpits over the Tibetan Plateau glaciers. Science of the Total Environment, 612, 1340-1347, DOI: 10.1016/j.scitotenv.2017.08.267 Parvin, F., Seki, O., Fujita, K., Iizuka, Y., Matoba, S., Ando, T., Sawada, K.(2019) Assessment for paleoclimatic utility of biomass burning tracers in SDome ice core, Greenland. Atmospheric Environment, 196, 86-94. DOI: 10.1016/j.atmosenv.2018.10.012 You, C., Yao, T., Xu, C. (2019) Environmental Significance of levoglucosan records in a central Tibetan ice core. Science Bulletin, 64, 2, 122-127, DOI: 10.1016/j.atmosenv.2018. Zennaro, P., Kehrwald, N., Marlon, J., Ruddiman, W.F., Brucher, T., Agostinelli, C., Dahl-Jensen, D., Zangrando, R.,Gambaro, A., Barbante, C. (2015) Europe on fire three thousand years ago: Arson or climate? Geophysical Research Letters, 42, 12, 5023- 5033, DOI: 10.1002/2015GL064259 Response: By taking the suggestion, we have added the following references through out the revised MS. Parvin et al., 2019; You et al., 2019; Grieman et al., 2017; 2018a,b; Li et al., 2018; Zennaro et al., 2018; and Legrand et al., 2016. Please see lines 58-61. Thank you for providing these references.

Page 3, lines 58-61: Do you mean that this is the first time that all three specific biomass burning tracers (levoglucosan, dehydroabietic and vanillic acid) were analyzed together in an ice core? Or do you mean that this is the first time that Kawamura et al. investigated these three markers. Response: We have deleted the term of "first" in revised MS. Please see line 91-93.

Page 4, Line 62: Please be clear as to which three compounds you are investigating in the paper. Response: By taking a comment, we wrote clearly about three compounds. In this paper we reports three compounds, that is, levoglucosan, dehydroabietic acid

and vanillic acid) in an ice core. Please see lines 91.

Page 4, Line 63 and continuing throughout the paper: Acronyms can tend to interrupt the flow of reading instead of helping. Use the name "Aurora Peak" throughout the rest of the paper instead of the acronym "APA." Response: Thank you, by taking the reviewer's comment we have used the name "Aurora Peak" throughout the paper. Please see the revised MS.

Page 4, Line 64 and continuing throughout the paper: All dates must include C.E. or A.D. Response: Thank you. By taking the reviewer's comment, we have added all dates are in A.D.

Page 4, Line 71 and continuing throughout the paper: Many of your references contain both the numbers from your citation software, followed by the written names of the references. Carefully check the document and omit all numbers related to the references. (In this case, you would change "Eastern Europe2 (Kawamura et al., 2012)" to "Eastern Europe (Kawamura et al., 2012)". Response: Thank you. We have corrected throughout the revised MS.

Page 4, Line 80: Here you cite Tsushima, 2014 and Tsushima et al., 2014. Your references state that both were published in 2015. Please double-check these dates. Response: Sorry for the mistake. We have corrected the dates, Tsushima, 2015 and Tsushima et al., 2015. Please see line 114.

Page 4, Line 86: Pokhrel et al., 2015b does not exist in your reference list. Instead, you cite a paper that was submitted to ES&T in 2016. In a search for this paper on "Web of Science" this paper was not published. It is not acceptable to use unpublished – and possibly rejected – work as a reference. Response: Thank you, we have deleted this reference in the revised MS.

Page 4, Lines 62-72: This paragraph ends abruptly. The authors state the aim of the study at the beginning of the paragraph, then dive into specifics, and then abruptly

stop. The study goal becomes lost in this paragraph structure. Please revise. Response: Thank you for the suggestion. By taking the reviewer's comment, we have added following lines in the revised MS. "This study covers 1665-2008 A.D. to better understand the historical atmospheric transport variability between the western North Pacific region (Kawamura et al., 2012) and eastern North Pacific region (this study) from source regions as well other ice core studies (e.g., Greenland, Svalbard and Akademii Nauk in the NH). The results of this study can further disclose the database of levoglucosan, dehydroabietic and vanillic acids from the alpine glacier in the North Pacific rim to explore their possible sources, origin, long and short-range atmospheric transport, ecological changes and climate variability in the NH. Please see lines 91-101 in the revised MS. Page 4, Lines 74-81: What evidence do you have that the ice core is 274 y BP at the depth of 180 m other than the annual layer counting? Cite Tsushima, 2015 a and b here. (Your citation of Tsushima, 2014 is incorrect as the papers you cite were published in 2015. Also, one of the Tsushima papers should be labeled "a" and the other should be "b"). In the Pokhrel, 2015 dissertation abstract, the age at 180 m is cited as 343 y BP. Of course, with new knowledge, depth-age scales can change. However, what DID change? Did you acquire independent dates? Response: Thank you, by taking reviewer's suggestion, we have cited Tshushima, 2015 and Tsushima et al., 2015. We did not acquire independent dates. Dates were measured by another group of our Institute of Low Temperature Science, Hokkaido University. Hence, we cited her paper. Please see line 114.

Page 4, Lines 80-81: Do you mean that 5-10 mm was shaved off the outside section of the core? Off all sections? How did you clean the ceramic knife? Response: Yes, we shaved off 5-10mm for all the sections. We cleaned ceramic knife 3 times by using organic free pure water (MiliQ water), 3 times by methanol (MeOH), 3 times by dichloromethane (DCM) and 3 times by mixture of 2:1 DCM and MeOH. Please see lines 117-122.

Page 4, Line 84: What kind of container? LDPE? Glass? What size? Were these

containers cleaned? If so, how? Response: We used IWAKI and/or PYREX's clean glass beaker of 1000 mL. First, we baked those containers for 12 hours at 450ïĆřC and then cleaned them three times by using organic free pure water (MiliQ water), 3 times by methanol (MeOH), 3 times by dichloromethane (DCM) and 3 times by mixture of 2:1 of DCM and MeOH, respectively. Please see lines 122-128.

Page 4, Line 85: What do you mean by a "standard clean room"? A Class 100? A Class 10,000? Response: Thank you. We just used a clean room (level-2). Please see line 124.

Page 4, Lines 87-88: Do you mean one quarter of the ice core by depth or by circumference? You mention that the sampling frequency was approximately 40% of the entire ice core. However, if you are taking 25% of the core, then how do you get a sampling frequency of 40%? If you are sampling by circumference, then you would have a continuous record. If you are sampling by depth, then you would have approximately 50% of the core. Are there locations in the core that you were not able to sample due to breakage, etc? If so, then note these locations. Response: Thank you for the question. We used one-quarter of the ice core by circumference. Only 40% of samples were available to measure for biomass burning (BB) tracers for our group. It does not have any special reasons. There was no breakage ice chronology. Please see lines 117, 128-129.

Page 4, Line 102: What are the "authentic standards" that you used? What concentrations? Where did you purchase the standards? Did you use any isotopically-marked standards? Did you include the standards in the samples as internal standards? Methods and materials: Do you have any lab blanks? Do you have any procedural blanks? If so, do you have any way for accounting for the concentrations of the three analytes in your blanks? If you do not have any blanks, what sort of QA/QC measures did you apply? What is the LOD for the analyzed compounds? Was each sample analyzed in triplicate (Page 5, Line 105)? Or were only three samples analyzed in triplicate? Response: Thank you very much for the comment. The authentic standard contains

levoglucosan (5.5 ng/ïĄ∎L), dehydroabietic acid (4.7 ng/ïĄ∎L), and vanillic acid (4.2 ng/ïĄ∎L), which was purchased from Wako, Japan. We did not use any isotopically marked standards. Instead, n-tridecane (C13H28) is used as an internal standard. We did not find levoglucosan, dehydroabietic acid, and vanillic acid in lab blank as well as procedural blanks. Only one sample was used for duplicate analysis. LOD for analyzed compounds 0.002 to 0.005 ng/kg-ice. Please see lines 145- 159.

Page 4, Line 104: You mention that analytical details are included in Simoneit et al., 2004, yet you do not include this paper in your references. A literature search demonstrates two options where Simoneit et al., 2004 investigates levoglucosan, etc. However, neither of the two possible Simoneit et al., 2004 papers nor Fu et al., 2008 (who you mention as another paper that contains the full method details) include essential information such as the m/z, the amount of time analyzing each m/z, etc. While you can still cite these papers, you still do need to include the primary information of the method. Response: By taking the reviewer's suggestion, we have added detail methods, including m/z value in the revised MS. Please see lines 130-159.

Page 6, Line 116: When you state an "important fraction", please mention how this fraction is important. Due to the volume? Due to the fact that anhydromonosccharides were produced by biomass burning? Response: Thank you, we have deleted the phrase in the revised MS.

Page 6, Lines 125-128: Here you argue that Figure 1 demonstrates that Aurora Peak is far away from any biomass burning sources, but in Figure 5 you demonstrate that there are fire sources near the peak. Explain this discrepancy. Response: We have deleted this line in the revised MS. To make a clear, we have added new figure of air mass back trajectories (BTs) together with fire counts. The results of BT and fire counts showed that Alaska was influenced by biomass burning from Siberia, Russia, Europe, China, Mongolia, Canada, and Japan. Please see Figure 7a-f and lines 160-173.

Page 6, Line 119: Here, you state that levoglucosan is only produced at temperatures

above 300ïC′ ĔĞr C, and cite references from 1984 until 2002. However, more recent literature (Kuo et al., 2011) states that levoglucosan and its isomers are only produced at temperatures up to 350ïC′ ĔĞr C. Explain this discrepancy. The citation for Kuo et al. is the same as you use later on the same page:

Kuo, L-J., Louchouarn, P., Herbert, B.E. (2011) Influence of combustion conditions on yields of solvent-extractable anhydrosugars and lignin phenols in chars: Implications for characterizations of biomass combustion residues. Chemosphere, 85, 797-805 doi:10.1016/j.chemosphere.2011.06.074

Response: Thank you for the comment; we have added the following lines in the revised MS. "Recently, Kuo et al. (2011) further reported that levoglucosan and its isomers produced at temperature up to 350ïĆřC." Please see lines 183-184.

Section 3.1: What evidence do you have that the fact that mannosan and galactosan are consistently below the limit of detection is due to these isomers not being present, versus to these isomers simply not being detectable by the analytical method? As mannosan is below the detection limit, the statement that "Thus, levoglucosan/mannosan mass ratios (L/M) could be relatively high" does not make sense. You would have to divide your levoglucosan results by zero. As you do not have quantifiable numbers for mannosan and glacactosan, the paragraph (Page 7, Lines 143-151) does not add value to the paper, and can be omitted. Response: Thank you for the question. We have deleted this paragraph in the revised MS because mannosan and glacactosan were not detected in this study. We used the same method reported by Kawamura et al. (2012), who already reported these species. We believe that this method is suitable to detect mannosan and galactosan.

Page 8, Lines 171-174: Many studies have wide ranges for the atmospheric lifetime of levoglucosan. The Hennigan et al., 2010 study is on the extremely low end of these calculations. Please include other studies and results to give a more accurate range. Response: Thank you. By taking a reviewer's comment, we have modified sentences

as follows. "Although levoglucosan may not be as stable as previously thought in the atmosphere (Hoffmann et al., 2010; Fraser and Lakshmanan, 2000), its concentrations are not seriously influenced during the transport for several days (Hoffmann et al., 2010; Mochida et al., 2010).

However, degradation of levoglucosan depends upon the levels of OH radicals (Hennigan et al., 2010), which are seriously affected by relative humidity of the atmosphere and air mass aging during long range atmospheric transport from East and North Asia to southern Alaska (Lai et al., 2014; Hoffman et al., 2010). We have added new references." Please see lines 211-218.

Page 8, Line 175 to Page 9, Line 189: (i) Do you mean that the source regions are southern Alaska as well as the possible source regions that are listed in lines 181-182? If so, then why do you separate southern Alaska? (ii) Do you mean that the heavy forest fires in eastern Siberia occur now, or occurred in the past, or both? (iii) Why do you compare your regional record to global biomass burning? (iv) Do you have a regional biomass burning record from charcoal or other data? (v) Do you have any indication of land-use change in eastern Siberia in the 1840s? Response: Thank you. (i) This time we performed back trajectory analysis together with fire counts. Hence, the sources are Siberia, East Asia, Europe, Canada, higher latitude of Alaska. Hence, we reorganized the lines. Please see lines 223-225. (ii) It occurred in the past (e.g., Ivanova et al., 2010). Please see line 225-228 in the revised MS. (iii) We also got at least nine higher peaks of levoglucosan (Figure 2a) during the same time as 1750s-1840s (Marlon et al., 2008). Hence, we compare to global biomass burning. (iv) We do not have charcoal data. Back trajectory analysis showed that Alaska was influenced from Europe, Russia, Siberia, East Asia, and Canada. Hence, we compare with global biomass burning. (v) Marlon et al. (2008) reported it. Hence, we have cited this reference in the revised MS. Please see line 229-231.

Page 9, Lines 190-200: This paragraph is illogical. Please omit the sentence "We did not detect significant concentrations of any isomers as we have discussed above" as

discussing isomer ratios does not fit into this paragraph. Response: By taking the reviewer's comment, we have deleted the sentences in the revised MS.

Section 3.1: (i) Are these spikes individual points or are they multiple points consecutively in the ice core? (ii) How can you determine if the spikes indicate a close fire, versus a fire that is farther away? (iii) Do the spikes indicate fire intensity? Response: Thank you for the questions. (i) These spikes are the individual point in the ice core. (ii) We cannot determine it directly. However, by using back trajectory analysis together with fire counts (2002-2007) we can indicates that long-range atmospheric transport is important in Alaska. (iii) Yes, spikes also may indicate fire intensity.

Page 9, Lines 201 to 204: The argument "These suggest that ice core $NH4+$ has common sources in the circumpolar regions" does not logically follow from the preceding sentence. Response: Thank you, we have deleted the ionic part in the revised MS.

Pages 9 and 10: The distances between Alaska and Greenland is thousands of kilometers. (i) Do you have other evidence than similar spikes of $NH4+$ between Mt Logan, GISP 2, and 20D to suggest "that ice core $HN4+$ has common sources in the circumpolar regions?" (ii) Are these spikes just visually the same, or is there some statistical test done to determine that these spikes correlate? (iii) Are the spikes just individual points, or are they peaks over decades? The following review paper investigates boreal fire source regions and the atmospheric transport, with implications for your assumptions on pages 9 and 10. Essentially, it would be difficult, although not impossible for the fire source regions to be the same for both Alaskan and Greenland ice core records: Legrand, M., McConnell, J., Fischer, H., Wolff, E.W., Preunkert, S., Arienzo, M., Chellman, N., Leuenberger, D., Maselli, O., Place, P., Sigl, M., Schuepbach, Flannigan, M. (2016) Boreal fire records in Northern Hemisphere ice cores: a review. Climate of the Past, 12, 2033-2059, doi:10.5194/cp-12-2033-2016.

Response: Thank you very much. The common source regions for these higher spikes could be East Asia, Eastern Russia, Siberia, higher latitudes of Alaskan regions, and

[Figure]

Canadian regions for those above-mentioned study sites. We assume that common source regions contributed the similar higher spikes for Mt. Logan, Greenland and Aurora Peak of Alaska.

We observed the similar higher spikes during the same periods for these sites. Except for the similar higher spikes, we do not have more evidence. The spikes are visually same. We did not perform the correlation analysis. Hence, we have added a few lines. Please see lines 259-260 and 283-288.

Section 3.1: Why is Whitlow et al., 1994 your primary reference when the research into boreal forest fires and ice core records has increased substantially in the past 25 years? Response: Thank you for the comments, this time we have added many new references. Please see the same section.

Pages 10 and 11: Lines 232-239: Would you like to say that the fires after 1900 only affect Mt. Logan and not Aurora Peak? Why is Aurora Peak more similar to Greenland records than to other Alaskan records? Please clarify. Response: Thank you very much. Yes, I would like to say that after 1900, fires affect more to Mt. Logan than Aurora Peak. In 19th century, the concentration of levoglucosan is drastically decreased in Alaska. Greenland and Alaska have common source regions of Siberia, Eastern Russia. Please see lines 297-301. Page 11, Lines 245-246: What mechanism do you propose for increased dilution and/or scavenging of biomass plumes after the 1830s? Would this mechanism affect the entire Arctic or just southern Alaska? Response: Thank you. "The mechanism could be dry and wet scavenging, diffusion, and degradation by hydroxyl radicals in the atmosphere during long-range atmospheric transport." Hence, we have added above lines to revised MS. Please see lines 291-295.

Page 11, Lines 245-247: In what ice core(s) do the see the difference in the concentrations? Is this a comparison between Aurora Peak and Greenland ice cores? If so, why would you expect a similarity over half a hemisphere of distance? Response: Thank you for the comment. We have modified the sentence as below in the revised MS. "Mt.

Logan, Svalbard, Tunu of Greenland and Aurora have common source regions, e.g. Russia and/or Siberian forest as well the North America/Canadian forest." Please see lines 297-301 and Figure 6a-e.

Page 11: Lines 247-249: You state "These special events further suggest that Alaskan glaciers cannot preserve most biomass burning events in the circumpolar regions, which occurred in the source regions of Siberia and North America". Do you mean that the combination of distance and atmospheric transport means that most fires in Siberia and North America will not be recorded in Alaska mountain glaciers? Do you mean that the difference between the Alaskan and Greenland ice core biomass burning records suggest that the Alaskan glaciers are not as good of receptors as the Greenland ice sheet? Response: We have deleted these lines in the revised MS.

Page 11, Lines 257-258: By mentioning the Little Ice Age, do you mean to imply that the cooler weather influenced the decreased biomass burning? Do you have any evidence or records of decreased temperatures in Siberia during this time period? Response: Thank you very much for the comment. We rephrased this sentence in the revised MS. Please see lines 465-473.

Page 13, line 287-289: Do you mean that the East Asian regions are more important for regional levoglucosan production or that they are more important as a source of levoglucosan for Aurora peak? You do go on to discuss this point further in the next few paragraphs, but as this is the first time that the reader is exposed to this idea, it is better to be clear from the onset. Response: Yes, correlation of levoglucosan with dehydroabietic and vanillic acids from 1920 to 1977 A.D. are not significant (ïĄť=0.11, 0.14) but vanillic vs. dehydroabietic acid showed significant correlation (0.41, p<0.01), suggesting different source region for levoglucosan. Please see lines 358-364.

Again, the significant correlation (Figure 3a-c) between dehydroabietic acid (except for 2005) and vanillic acid (ïĄť= 0.60, p<0.01) is better than the correlations of levoglucosan with dehydroabietic (except for 1981 and 1986) acid (0.30) and vanillic acid

(0.21) after the Great Pacific Climate Shift (GPCS) that is, 1977-2007 AD. Please see line 420-423 and Figs. 3.

Backward trajectories analysis (500 hPa) of air masses (2002-2007 AD) together with fire counts, also showed that source regions are also Mongolia, China and Japan (Figure 7a-f).

Pages 14 and 15: Why would the long-range atmospheric transport be insignificant? It may just be that regional transport overwhelms the long-range signal from the 1920s until the present, but the same amount of long-range transport may occur. What do you mean that the concentrations "are secure"? Response: Correlation of levoglucosan with dehydroabietic and vanillic acid from 1920 to 1977 A.D. are not significant (Kendall's ïĄť=0.11, 0.14) but vanillic vs. dehydroabietic acid showed significant correlation (0.41, p<0.01), suggesting different source region for levoglucosan. Please see lines 358-364.

Again, the significant correlation (Figure 3a-c) between dehydroabietic acid and vanillic acid ïĂíïĄť= 0.60, p<0.01) is strong than those of insignificant correlations of levoglucosan with dehydroabietic acid (0.30) and vanillic acid (0.21) after the Great Pacific Climate Shift (GPCS) that is, 1977-2007 AD. Please see line 420-423.

In addition, from 1980s to onwards, trends of dehydroabeitic acid is increasing (similar to Kamchatka ice core records) and concentration is higher than levoglucosan. These results strongly suggest that they have different sources. Hence, for the dehydoabeitic acid and vanillic regional transport overwhelms the long range atmospheric transport. Hence, we added following lines to revised MS. "….regional transport overwhelms the long-range atmospheric transport….." Please see lines 383-385.

We have deleted the word "secure" in the revised MS.

Pages 14 and 15: Why did you choose to do a point to point correlation of data from a time series? The data are definitely skewed, with the majority of the data with low

concentrations and then a few separate spikes. Why did you not choose other types of correlations that may be more applicable? What is the statistical level of correlation of each of these factors? Response: Yes, we agreed with reviewer's comment. By taking the comment, we have added non parametric Kendall's correlation (ïĄť) and explained the results accordingly. Please see Figures 3 and 4.

Page 17, Line 400: In what way do the forest fire signal depend on the source region? Do they depend on proximity to the source? Do they depend on the type of vegetation burned in the source region? Response: Thank you very much. Yes, the forest fire signals depend on the source region and proximity to the source, and types of vegetation. Hence, we have added one sentence in the revised MS. Please see lines 561-566.

Page 17, Lines 401-403: Your paragraph is better without this sentence. Please remove. Response: Thank you. We have deleted the lines.

Figure 5: Why do you investigate the MODIS fire spots over different areas for each year? For example, the 2004 plot stops at âĹij45 degrees N, while the 2006 plot stops at âĹij 32 degrees N. Why do you include much of the United States, if regions south of 45 degrees north are unlikely to be a source region? If Siberia is a major source region, why did you not also investigate Siberia with any available data? Response: Thank you. By taking comments, we have added MODIS fore spots for major source region together with cluster analysis. Please see Figure 7a-f.

TECHNICAL CORRECTIONS: Title: Place "the" before "1660s" in the title Response: Corrected. Please see title.

Page 2, Line 18: Change "melt" to "melted" Response: This sentence was deleted.

Page 2, Lines 24 and 25: Change "there are few discrepancies of higher spikes among them after the 1970s" to "there are a few discrepancies of higher spikes especially after the 1970s". Response: The sentences were deleted.

Page 1, Line 27: Place "as well as" before "other higher plants" Response: This sentence was deleted.

Page 2, Line 29: Change "regions of southern Alaska, being different from previous ice core studies" to "regions of southern Alaska. These results differ from previous ice core studies." Response: The phrase was deleted.

Page 2, Line 34 and Page 17, Line 398: The word "gleaming" is wonderful, and I am sorry to suggest replacing this word with more boring options. Unfortunately, it is not quite clear what you would like to suggest with this word. Do you mean substantial? Clear? Definitive? If so, please use one of these words. Response: These words are deleted.

Page 3, Line 39: Replace "provide the" with "archive" Response: Corrected. Please see line 51.

Page 3, Line 42: Omit "which are reported elsewhere" Response: Omitted.

Page 3, Lines 45-46: with "have some extent on climate change effect" do you mean to say "may affect climate change"? If so, then replace the phrase. Response: Corrected. Please see line 49.

Page 3, Line 54: Place "a" before "pyrolysis" Page 4, lines 66-67: Change "Particularly, 10 day backward trajectory" to "Particularly, 10-day back trajectories" Response: These words are deleted.

Page 4, Lines 85-86: By "All steps are followed as reported previously prior to analysis (Pokhrel, 2015; Pokhrel et al., 2015b)" do you mean "All analytical steps are previously reported in Pokhrel, 2015 and Pokhrel et al., 2015b"? Response: We added references of Fu et al., 2011 and Kawamura et al., 2012 with some explanation in the revised MS. Please see section 2.

Page 5, Line 89: Change "melt" to "melted" Page 5, Line 90: Change "shape" to "shaped" Response: Corrected. Please see lines 130 and 131.

[Figure]

Page 5, Line 93: Choose to use either chemical formulas or names, and then be consistent throughout the paper. Response: Corrected throughout the section 2.

Page 5, Line 94: Change "reported previously" to "previously reported" Response: Deleted these words in the revised MS.

Page 5, Line 107: Change "were" to "was" Page 5, Line 108: Change "were" to "was" Response: Deleted.

Page 5, Line 108: Omit "traject" before "compounds" Response: Corrected to "target". Please see line 156.

Page 5, Line 109: I am sorry, but I do not understand what you would like to say by "physical functioning fire smoldering spot". Please change this phrase. Response: Deleted these words in the revised MS.

Page 6, Line 116: Place "to" before "an important fraction". Response: Deleted this line in the revised MS.

Page 6, Line 125: Omit "(i.e., distribution)" Response: It is removed.

Page 6, Line 128: Omit "the" before "Aurora Peak" and replace "the" with "any" before "biomass burning" Page 7, Line 144: Change "sifnificant" to "significant" Response: Thank you. Both are deleted.

Page 7, Line 160: This is the first time that you have used the acronym "BB". Replace with "biomass burning" and do not use the acronym. Response: Thank you, deleted.

Page 8, Line 162: Replace "didn't" with "did not". Response: Deleted.

Page 8, Line 175: Replace "around" with "the" Response: Deleted.

Page 9, Line 192: Use the full word "levoglucosan" rather than "Lev". Response: Thank you. Corrected throughout the MS.

Page 8, Line 183 and Page 9, Line 198: Replace "borel" with "boreal" Page 9, Line

204: Replace "got" with "obtained" Response: Thank you. We have corrected them and deleted this word: got. Please see lines 226, 246, 357.

Page 10, Line 212: Change "is consistent to" to "is consistent with" Page 11, Line 238: Replace "only the exception" with "the only exception" Response: Thank you very much. Please see lines 254 and 281 in the revised MS.

Page 11, Line 240: Replace "Above results and discussion suggest the subsequent evidences" with "The above results suggest" Response: Corrected. Please see line 289 in the revised MS.

Page 11, Line 242: Replace "souhtern" with "southern" Response: Corrected. Please see line 291.

Page 11, Line 251: Unfortunately, "heavy" does not describe forest fires. Do you mean intense? Do you mean widespread? Response: Thank you. I mean both, intense and widespread. I have added both words in the revised MS. Please see line 304.

Page 11, Line 249: Replace "Siberian" with "Siberia" Response: Corrected.

Page 11: Line 248: Replace "can not" with "cannot" Response: Deleted.

Page 11, Line 254: Replace "declined" with "declining" Response: Corrected. Please see line 308.

Page 12, Lines 280-283: Change to "These results suggest that biomass burning plumes of pine, larch, spruce and fir trees in Siberian regions (Kawamura et al., 2012: Ivanova et al., 2010) have a substantially larger influence on Kamchatka, southeastern Russia than on southern Alaska". Response: Thank you very much. We have replaced it. Please see lines 347-350.

Page 12, Line 286: Change "borel" to "boreal" Response: Deleted.

Page 13, Line 296: Change "discrepancy" to "discrepancies" Response: Corrected. Please see line 371.

Page 13, Lines 297 to 298: Change "Kamchatka showed gradual increase after the 1950s" to "Dehydroabietic acid concentrations gradually increased in the Kamchatka ice core after the 1950s". Response: Corrected. Please see lines 372-373.

Page 13, Line 305: Change "doesn't" to "does not". Page 14, Line 326: Change "conifer rich" to "conifer-rich" Response: Deleted this phrase. Corrected, Please see line 406 in the revised MS.

Page 15, Line 347: Remove "from" before "climate driven" Response: This line is deleted in the revised MS.

Page 15, Line 349: Change "could be" to "may be" Page 16, Line 369: Replace "constitute" with "constituent" Response: These words are deleted in the revised MS.

Page 16, Lines 377-384: You only use the acronym "NPR" three times. It is much better to use the words "North Pacific Rim" than an acronym that introduces confusion. Please use the words for this phrase rather than the acronym throughout the paragraph. Response: Corrected throughout the revised MS.

Page 17, Line 395: Replace "with early study" with "than a previous study" and then cite the study in the sentence. Figure 1: Place "a" before "180-meter". This figure does not need a citation unless you are using the exact figure as in your earlier work. Response: Thank you, we have deleted it. We place "a" before 180-meter. Please see the figure caption. We have modified Figure 1 in the revised MS.

---

## Author Comment (AC2) · 12 Sep 2019

**Reviewer 2 "Ice core records of biomass burning tracers (levoglucosan, dehydroabietic and vanillic acids) from Aurora Peak in Alaska since 1660s: A new dimension of forest fire activities in the Northern Hemisphere" by Ambarish Pokhrel and co-workers. Overall evaluation:**

The topics is of importance since fires are a major source of gases and aerosols that

strongly impact chemical composition of the atmosphere and the radiation balance. In turn, climate changes directly disturb the fire regime, for instance through the duration of fire weather conditions and changes in vegetation, particularly in the boreal regions. In addition to this overall interest, the data presented in this paper would be useful to discuss the consistency between these three organic markers (levoglucosan, dehydroabietic and vanillic acids) and other potential proxy including ammonium. However, as it stands, the paper suffers from too many weaknesses to be recommended for publication at the ACP journal. I recommend to the authors to take time to revisit the existing literature and their data. Major weaknesses:

1. Quality of the Aurora ice record: Some key information are missed in the manuscript for the reader to evaluate the quality Aurora ice core record. Indeed, when using an ice core record to infer atmospheric information, the reader (and the reviewers) needs to have some basic information that are not given in section 2 (Materials and Methods). I think that, given the rather low elevation (2850 m) of the Aurora site, we may expect frequent melting. If, so that has to be clearly stated in the manuscript and the authors would discuss the possible consequence for the quality of the ice record in terms of atmospheric signal. Since the effect of melting is not well know for organics, it would be nice to show the record of major ions (including ammonium, nitrate, and sulfate). Checking your Figure 4, i am very surprised by the nitrate levels that are shown to range between 0 and 34 ppb (i.e., very low levels). At the opposite, the nss-K level (ranging from 0 to around 50 ppb) exhibits several values exceeding 15 ppb (which is a lot): how much abundant is calcium in this core? (see my further comments on the use of fine potassium). Responsse: Thank you very much. The annual snow accumulation rate is 8 mm yr-1 since 1900 to onwards and drastically accumulated at a rate of 23 mm yr-1 after the Great Pacific Climate Shift. Meanwhile, the average temperature anomalies for 1923-1942, 1943-1945, and 1976-2007 were +0.73, -0.65 and +0.55C, respectively (Tsushima et al., 2015).

We assumed that there was not 100 percent melting of snowfall in the saddle of the

[Figure]

Aurora. The correlations of D record of this ice core and detrended annual accumulation rates of snowfall are well correlated with air temperature and precipitation amount of Aurora.

The average annual amplitude of D from this ice core is 30.9%. This high amplitude (more than 30%) cannot be maintained, if there was intensive melting (100%) in the past (Tsushima et al., 2015). Please see lines 54136-54550.

We are preparing a separate manuscript of ions. We are very sorry to say that we would like to remove Fig. 4 and explanation of this figure. Hence, this time we have deleted these two paragraphs and Fig.4.

2. Inconsistencies: Since you will discuss in section 3 (as also mentioned in the abstract) the correlations of levoglucosan with $NO2-$, $NO3-$, $nss-SO42-$, $nss-K+$, and $NH4+$ that are all insignificant (suggesting that these anions and cations do not represent a gleaming signal of biomass burning activities in the source regions for southern Alaska)), it would be nice to show the profiles. This need to report these profiles also comes from the fact this observed absence of correlations contrasts with the statement that I find in the paper from Tsushima et al. (2005) stating "To confirm the dating based on D and $Na+$ seasonal cycles, we compared the dating of the ice core with reference horizons of known age (Fig. 3). We found a large peak of $NO3$ and $NH4+$ and a visible dirty layer at 8.55 m w.eq., which we ascribed to the year 2004. Generally, $NO3$ and $NH4+$ are released by forest fires (e.g., Legrand and Mayewski, 1997; Eichler et al., 2011). Âż This point clearly needs to be discussed showing all the records, i think. Since, as also suggested by your figure 5, the 2004 year was characterized by large fires in Alaska, please also comment your organic records for this year ? Response: Thank you very much. This time we have removed that ionic part in the revised MS. We are preparing another paper for ions.

We did not obtain high value of levoglucosan (95.3 ng/Lkg-ice) compared to its average (543). But dehydroabietic (309.7) and vanillic (2.70) acids showed higher concentrations compared to their average (62 and 1.6 ng/kgL-ice, respectively). Hence, there is more local influence for dehydroabeitic and vanillic acids. Please also see Figure 6e in the revised MS.

3. Numerous previous works are not cited or adequately cited in the manuscript: In the introduction and at several places in the text, the previous works done on fire records in ice cores are not adequately cited, and some important references are missed including two reviews papers (see the list below). For example, you extensively cited the paper from Whitlow et al. (1994) for ammonium and nitrate biomass burning events that just follows the pioneering study from Legrand et al. (1992) for ammonium, nitrate and carboxylates. After the publication of these two papers, it becomes clear that, although some ammonium spikes are sometimes accompanied by nitrate peaks, it is not a general rule (Savarino and Legrand, 1998). This point was extensively discussed in the review from Legrand et al. (2016). The same is true for the non-sea-salt and non-dust potassium fraction. On this topic, in your manuscript I would recommend to report nss-non-dust-potassium (calculated by using your calcium data). Finally, none of the Greenland ice core studies reported a sulfate perturbation with biomass burning peaks. So I will be more careful about that at line 246. Response: Thank you very much. By taking the reviewer's comment, we have cited all these research papers in the revised MS.

We are preparing another paper by focusing on ions. Nss-non-dust K+ calculation is of great interest. We will certainly report nss-non-dust K fraction in another paper. Hence, we have decided to delete all the ionic parts of this ice core on thisfrom this MS.

Legrand M., M. De Angelis, T. Staffelbach, A. Neftel, and B. Stauffer, Large perturbations of ammonium and organic acids content in the Summit Greenland ice core, fingerprint from forest fires ?, Geophys. Res. Lett., 19, 473-475, 1992. Legrand M., and M. De Angelis, Light carboxylic acids in Greenland ice: A record of past forest fires and vegetation emissions from the boreal zone, J. Geophys. Res., 101, 4129-4145, 1996. Savarino, J., and M. Legrand, High northern latitude forest fires and
* * *
Interactive
comment

vegetation emissions over the last millenium inferred from the chemistry of a central Greenland ice core, J. Geophys. Res., 103, 8267-8279, 1998. Legrand, M., McConnell, J., Fischer, H., Wolff, E. W., Preunkert, S., Arienzo, M., Chellman, N., Leuenberger, D., Maselli, D., Place, P., Sigl, M., Schüpbach, S., and Flannigan, M.: Boreal fire records in Northern Hemisphere ice cores: A review, Clim. Past, 12, 2033-2059, doi:10.5194/cp-12-2033-2016, 2016. Rubino, M., D'Onofrio, A., Seki, O., and Bendle, J.A., Ice-core records of biomass burning, The Anthropocene Review, vol. 3(2), 140-162, DOI: 10.1177/2053019615605117, 2016. Grieman, M. M., Aydin, M., Isaksson, E., Schwikowski, M., and Saltzman, E. S.:Aromatic acids in an Arctic ice core from Svalbard: a proxy record of biomass burning, Clim. Past, 14, 637-651, https://doi.org/10.5194/cp-14-637-2018, 2018.

4. The wording is sometimes too vague or unclear, figures are unclear: I don't understand the meaning of "A new dimension of forest fire activities in the Northern Hemisphere" in the title. Response: Thank you, we have changed the title. Please see lines 1-4 in the revised MS.

Line 29: what do you mean with "different ice core studies": Please specify from where ?, with which proxy ?, which time period ? Response: Thank you for the question. We have improved this line and mentioned clearly Kamchatka (Kawamura et al., 2012), Mt. Logan (Robock et al., 1991), Svalbard, Akademii Nauk and Tunu (Grieman et al., 2018a,b; 2017), and Greenland (Whitlow et al., 1994) ice core studies in the revised MS. Please see Figures 6 (a-e). Please see lines 34-36.

Lines 236 and 239: what are these numbers: concentrations of ammonium ? of levoglucosan ? Why figure 3 reports levoglucosan only up to 600 ng L-1 and Figure 4 only up to 1200 ng L-1 while Figure 2 indicates levo as high as 20802 ng L-1 ? Please show satellite data for Siberia as well in Figure 5. Response: Thank you. Figure 3 is the correlation analysis for the period of 1977-2007 (i.e., after the Great Pacific Climate Shift). Concentrations of these compounds are within 600 ng/kgL-ice. We can see in the figure caption too. We have prepared air mas backward trajectories (500 hPa) in

the revised MS. Please Figure 7a-f. Please see lines 160-173.

5. Information derived from Back-ward trajectories: I think you may address more details on the origin of air mass reaching the Aurora site. In the present manuscript you stated at the end of the introduction: "Particularly, 10 day backward trajectory from 1992-2002 showed that southern Alaska can receive air masses from the North Pacific Regions, East Asia, Eastern Russia-Siberia, higher latitudes of Alaskan regions, Japan, and Canadian regions in the troposphere (>300 hPa) (Yasunari and Yamazaki, 2009)." I here recommend to address the following points (that would need new calculations): (1) focus on the fire season (from June to August), (2) I think 300 hPa is too high (it is the upper troposphere) and 500 hPa (around 5 km elevation) is likely more relevant for the travel of plumes. Also check the sensitivity between 5 and 10 days. Response: Thank you. By taking reviewer comments, we have added Air mass Backward trajectories (500 hPa) with fire counts for the fire seasons whole year. (June to August). Please see Figure 7a-f.

6. Discussion with previous records (section 3.4): Why do you extensively discuss your organics with ammonium records from 20D (Greenland) ? This discussion is not very useful since the records were obtained with different proxy and are expected to be influenced by different source regions (Canada for Greenland versus Alaska and may be Siberia for Aurora). Instead, again please show your own (Aurora) data on ammonium, nitrate, potassium, sulfate etc. Response: Thank you. We have deleted ionic parts from this section of 3.4 (i.e., comparison with ammonium, nitrite, nitrate and sulphate of same ice core) as mentioned above.

This time, we compared this study with other ice core studies of biomass burning tracers in the section 3.4 (i.e., Biomass burning tracers, Temperature, and Climate variability: Atmospheric consequences) in the revised MS. Please see section 3.4.

---

## Author Comment (AC3) · 12 Sep 2019

Reviewers #3 Interactive comment on "Ice core records of biomass burning tracers (levoglucosan, dehydroabietic and vanillic acids) from Aurora Peak in Alaska since 1660s: A new dimension of forest fire activities in the Northern Hemisphere" by Ambarish Pokhrel et al.

 This manuscript presents records of the

biomass burning tracers levoglucosan, dehydroabietic acid and vanillic acid from an ice core retrieved at Aurora Peak, Alaska, and covering the time period from ca. 1660 to 2009. In general, this seems to be a high quality data set, which may be interesting and may deserve publication, since only few ice core records of such tracers are available up to now. Unfortunately, the manuscript does not meet basic scientific criteria, as outlined below, is very descriptive and not well-written (requires English editing), and lacks a clear structure, which makes it hard to digest. Before becoming publishable, major revisions are therefore required.

1. Method description is incomplete (no detection limits) and basic ice core data are missing (dating, dating uncertainty, melt extend, etc.). Response: Thank you. Taking reviewer comment, we have added several paragraphs on methods in this section 2. Please see revised section 2, lines 104-174.

2. The record presented is incomplete, only 40% of the core was analysed, i.e. the records are not suitable to discuss short-term biomass burning events. Response: Thank you very much. Only 40% samples were available for analyzing organic compounds (e.g., anhydrosugars). Based upon the available data, we have discussed the source of biomass burning tracers as best as we could.

3. Records should not be shown as continuous line; data points should indicate for which time period they are representative. Response: Thank you. We have added new figure in the revised MS. Please see Figure 2.

4. The Aurora ice core is affected by melting with melt feature percentages of up to 100%. It should be discussed how this effects the records of biomass burning tracers. Response: Thank you very much. The annual snow accumulations rate is 8 mm yr-1 since 1900 to onwards and drastically increasedincrease to 23 mm yr-1 after the Great Pacific Climate Shift. Meanwhile, the average temperature anomalies for 1923-1942, 1943-1945, and 1976-2007 were +0.73, -0.65 and +0.55ïČřC, respectively (Tsushima et al., 2015). Please see line 536.

We assumed that there was not 100 percent melting of snowfall in the saddle of the Aurora. The correlations of ïĄďD record of this ice core and detrended annual accumulation rates of snowfall are well correlated with air temperature and precipitation amount at Aurora. The average annual amplitude of ïĄďD from this ice core is 30.9%. This high amplitude of 30% cannot be maintained if a higher percentage (e.g., 100%) of the melting occurred in the past (Tsushima et al., 2015; Tsushima, 2015). These points are added in the revised MS. Please see lines 114113-117 116 and 541536-550545.

5. Recent other publications in this field should be discussed and cited (list at the end of this review). Response: Thank you. We have used new potential papers throughout out the MS. Please see revised MS.

6. There is no discussion with respect to other available data. Concentrations in the Aurora core seem to be higher than in the Kamchatka core, although Aurora is located much further away from the sources. Response: We compared our data with several other ice core studies in the revised MS. Please see Figure 6a-e.

7. What is the reasoning behind conducting a correlation analysis with nss-sulphate and nss-calcium. They are not expected to have a biomass burning source. Where the ion records averaged to match the incomplete sampling of the organic tracers? Response: Thank you. We have deleted these paragraphs.

8. It is unclear what can be learned from the fire spot data. Here you need to come up with a quantitative number to compare with ice core records. Response: We have prepared new fire counts with fire intensity and air mass backward trajectories. Please see Figure 7(a-f).

9. There are no conclusions, just a summary. Response: Thank you, this time we have added new paragraphs in this section. Please see the summary and conclusions in the revised MS.

10. Fig. 1: Already shown in Pokhrel et al., 2015 and Pokhrel et al., 2016, is this not a

copyright issue? 10. Response: Thank you. We have changed it. Please see Figure 1.

Thank you very much for your valuable time and comments.

Gambaro, A., et al. (2008). "Direct Determination of Levoglucosan at the Picogram per Milliliter Level in Antarctic Ice by High-Performance Liquid Chromatography/Electrospray Ionization Triple Quadrupole Mass Spectrometry." Analytical Chemistry 80(5): 1649-1655. Grieman, M. M., et al. (2017). "Aromatic acids in a Eurasian Arctic ice core: a 2600- year proxy record of biomass burning." Clim. Past 13(4): 395-410. Grieman, M. M., et al. (2018). "Aromatic acids in an Arctic ice core from Svalbard: a proxy record of biomass burning." Clim. Past 14(5): 637-651. Grieman, M. M., et al. (2018) "Burning-derived vanillic acid in an Arctic ice core from Tunu, northeastern Greenland." Clim. Past 14(11): 1625-1637. Grieman, M. M., et al. (2015). "A method for analysis of vanillic acid in polar ice cores." Clim. Past 11(2): 227-232. Kehrwald, N., et al. (2012). "Levoglucosan as a specific marker of fire events in Greenland snow." Tellus B: Chemical and Physical Meteorology 64(1): 18196. Zennaro, P., et al. (2014). "Fire in ice: two millennia of boreal forest fire history from the Greenland NEEM ice core." Clim. Past 10(5): 1905-1924.